

# AdaHRBF v1.0: Gradient-Adaptive Hermite-Birkhoff Radial Basis Function Interpolants for Three-dimensional Stratigraphic Implicit Modeling

Baoyi Zhang[1], Linze Du[1], Umair Khan[2], Yongqiang Tong[1,3], Lifang Wang[3,4], Hao Deng[1*]

[1]Key laboratory of Metallogenic Prediction of Nonferrous Metals & Geological Environment Monitoring (Ministry of Education) / School of Geosciences & Info-Physics, Central South University, Changsha 410083, China

[2]Institute of Deep-Sea Science and Engineering, Chinese Academy of Sciences, Sanya 572000, China

[3]Wuhan ZGIS Science & Technology Co. Ltd., Wuhan 430074, China

[4]School of Geomatics and Geography, Hunan Vocational College of Engineering, Changsha 410151, China

*Correspondence to*: Hao Deng (hoedeng@163.com)

**Abstract.** Three-dimensional (3D) stratigraphic modelling is capable of modeling the shape, topology, and other properties of strata in a digitalized manner. The implicit modeling approach is becoming the mainstream approach for 3D stratigraphic modelling, which incorporates both the off-contact attitudes and the on-contact occurrence information of stratigraphic

interface to estimate the stratigraphic potential field (SPF) to represent the 3D architectures of strata. However, the magnitudes of SPF gradient controlling variation trend of SPF values cannot be directly derived from the known stratigraphic attribute or attitude data. In this paper, we propose an Hermite-Birkhoff radial basis function (HRBF) formulation, AdaHRBF, with an adaptive gradient magnitude for continuous 3D SPF modeling of multiple stratigraphic interfaces. In the linear system of HRBF interpolant constrained by the scattered on-contact attribute points and off-contact attitude points of a set of strata in

3D space, we add a novel optimizing term to iteratively obtain the true gradient magnitude. The case study shows that the HRBF interpolants can consistently establish accurate multiple stratigraphic interfaces and fully express the internal stratigraphic attribute and attitude. To ensure harmony of the variation of stratigraphic thickness, we adopt the relative burial depth of stratigraphic interface to the Quaternary as the SPF attribute value and propose a new stratigraphical thickness index (STI) to represent the variation trend of stratigraphic thickness in SPF. In addition, the proposed stratigraphic potential field

modeling by HRBF interpolants can provide a suitable basic model for subsequent geosciences numerical simulation.

## 1 Introduction

The three-dimensional (3D) stratigraphic modeling and visualization technology is of great significance for the intelligent



management of subsurface space (e.g., mineral resource assessment, reservoir characterization, groundwater management, and urban subsurface space planning) and has garnered extensive attention from geologists (Houlding, 1994; Mallet, 2002). The two main methodologies of 3D stratigraphic modeling are so-called explicit and implicit modeling (Lajaunie et al., 1997). Traditional explicit modeling can be described as a modeling method of 3D geological boundaries that relies heavily on a complicated and time-consuming process of human-computer interaction for connecting the geological boundary lines to form a 3D model of geological surfaces, and it is difficult to update the model. Implicit modeling defines a continuous 3D stratigraphic potential field (SPF) that describes the stratigraphic distribution and represents geological boundaries using an implicit mathematical function. The increasing significance of implicit method in stratigraphic modeling stems from not only the advantages of speed, reproducibility and topological consistency over the traditional explicit modeling method but also the full representation of stratigraphic structure through SPF. Three-dimensional stratigraphic potential field modelling is to implicitly represent the nature, shape, topology, and internal property of a given set of strata. The stratigraphic interface is expressed by a specific equipotential surface of the SPF. Meanwhile, an implicit field function is capable of combining with geophysical fields (Lindsay et al., 2012; Wellmann et al., 2017; De La Varga et al., 2019), geochemical fields (Vollgger et al., 2015; Zhang et al., 2018; Wang et al., 2019) and using uncertainty analysis (De La Varga et al., 2019; Fouedjio et al., 2021), finite-element methods (Caumon et al., 2013), and forward or inversion methods (Grose et al., 2018; De La Varga et al., 2019) for specific geological analysis, e.g., structural analysis (Hillier et al., 2013; Basson et al., 2016; Laurent et al., 2016; Basson et al., 2017; Creus et al., 2018; Grose et al., 2018), metallogenic analysis (Basson et al., 2017; Zhang et al., 2018; Li et al., 2019; Wang et al., 2019), groundwater management (Zhang et al., 2022a; Hassen et al., 2016), and reservoir characterization (Khan et al., 2021; Zhang et al., 2022b; Ali et al., 2021). Therefore, using SPF to express a set of conformable strata and their attribute distribution in 3D space is convenient for spatial analysis, statistics, and simulation.

The attitude information can be incorporated into implicit modeling in HRBF method by setting up the gradients of implicit function. To control the attitude of the modeled strata, the attitude information (i.e., dip and strike directions) is encoded as the gradient directions. However, existing HRBF method constructs implicit field functions separately for each geological interface and extract the zero value equipotential surfaces to locate the geological interface. Therefore, it is difficult to maintain topological and semantic consistency between geological bodies. For modeling multiple strata in an integrated and unique framework, however, setting up the gradient magnitudes being adaptive to the attitude and thickness variations of strata is rather challenging. Assigning the adaptive gradient magnitudes to HRBF interpolant function is a "chicken-and-egg" problem: while the implicit function results from the gradients, the suitable gradient magnitudes are estimated from the reasonable implicit function.

In this study, we propose an Hermite-Birkhoff radial basis function (HRBF) framework for SPF modeling, AdaHRBF, which simulate multiple interfaces among a set of conformable strata by a unified one-step process. In this linear system of HRBF interpolant, we add a novel optimizing term to iteratively obtain the true gradient magnitudes. The particular case where the





SPF was reconstructed from geological maps and cross-sections demonstrates the advantages and general performance of stratigraphic potential field modeling using the AdaHRBF method. The SPF attribute value is set to the relative burial depth of strata, i.e., mean distance from a given stratigraphic surface to the top surface of the Quaternary, meanwhile, we propose a new stratigraphic thickness index (STI) to express the variation trend of stratigraphic thickness. The distributions of attribute, thickness, and attitude of strata in 3D space can be fully expressed by the SPF and its gradient vector field.

## 2 Related Works


The key of implicit modeling methods is to interpolate a 3D scalar field function whose equipotential surfaces indicate the boundaries of geological bodies. These surfaces can represent ore grade boundaries or stratigraphic interfaces. This scalar field is interpolated from stratigraphic interface points and attitude data with either discrete interpolation schemes or continuous interpolation schemes.

For discrete interpolation schemes of implicit modeling with a special mesh, Mallet (1989); Mallet (1992) proposed a discrete smooth interpolation (DSI) method of producing values only at the mesh points on the stratigraphic interface instead of explicitly computing a function defined everywhere. The GoCAD (www.pdgm.com/products/skua-gocad/) software was developed based on the DSI method to meet the needs of geological, geophysical, and petroleum reservoir engineering modeling (Collon et al., 2015; Zhang et al., 2020). Caumon et al. (2013) proposed a discretizing finite-element method

(FEM) to generate 3D models of horizons on a tetrahedral mesh, using stratigraphic interface traces of unknown attribute values and attitude measurements from 2D geological maps, remote sensing images, and digital elevation models. Hillier et al. (2013) presented a structural field interpolation (SFI) algorithm using an anisotropic inverse distance weighted (IDW) interpolation scheme derived from eigen analysis of strike/dip measurements. Gonçalves et al. (2017) proposed a vector potential-field solution from a machine learning perspective, recasting the problem as multivariate classification in a

compositional data framework, which alleviates some of the assumptions of the cokriging method.

Since the continuous interpolation schemes does not depend on a mesh for its definition, the stratigraphic interfaces can be extracted at any desired resolution in the specific volume of interest. There is already a dual kriging or cokriging formulation for continuous potential field modeling of multiple stratigraphic interfaces. Lajaunie et al. (1997) proposed an implicit potential field modeling method using the dual formulation of kriging interpolation that considers known points on a geological

interface and plane attitude data such as stratification or foliation planes. Calcagno et al. (2008) cokriged the location of geological interfaces and attitude data from a structural field to interpolate a continuous 3D potential-field scalar function describing the geometry of geological bodies. Geomodeller 3D (www.geomodeller.com), an implicit geological modeling application, utilizes the implicit potential field method by cokriging or the dual formulation of kriging (Lindsay et al., 2012; Hassen et al., 2016). De La Varga et al. (2019) presented GemPy (https://github.com/cgre-aachen/gempy), an open-source



implementation, to generate 3D geological models based on an implicit potential-field cokriging interpolation approach and to

enable stochastic geological modeling and inversions of gravity and topology in machine-learning and Bayesian inference

frameworks. Renaudeau et al. (2019) proposed an implicit structural modeling method using locally defined moving least

squares shape functions and solved a sparse sampling problem without relying on a complex mesh. Irakarama et al. (2020)

introduced a new method for implicit structural modeling by regularization operators using finite differences. To reduce the

impact of regularly occurring modeling artifacts that result from data configuration and uncertainty, Von Harten et al. (2021)

proposed an approach that is a combination of an implicit interpolation algorithm with a local smoothing method based on the

concepts of nugget effect and filtered kriging known from conventional geostatistics.

For continuous radial basis function (RBF) or HRBF interpolation schemes of implicit modeling without a mesh, Cowan et

al. (2003) constructed an implicit model of the orebody or stratigraphic interface using a volumetric RBF interpolation function

with an equipotential surface that includes the interface points, and conventionally assigned an attribute value of zero and a

"±" sign to indicate the inside and outside of the interface. Hillier et al. (2014) presented a generalized interpolation framework

using RBF to implicitly model 3D continuous geological interfaces from on-surface points with gradient constraints as defined

by strike-dip data with assigned polarity. Leapfrog Geo (www.leapfrog3d.com) is an implicit geological modeling software

package that models scattered data for interfaces using fast RBF interpolation methods (Vollgger et al., 2015; Stoch et al.,

2020; Creus et al., 2018; Basson et al., 2016; Basson et al., 2017). Martin and Boisvert (2017) developed a RBF-based

implicit modeling framework using domain decomposition to locally vary orientations and magnitudes of anisotropy for

geological boundary models. Zhong et al. (2019); Zhong et al. (2021) introduced combination constraints for modeling ore

bodies based on multiple implicit fields interpolation through RBF methods, in which a multiply labeled implicit function was

defined that combines different implicit sub-fields by the combination operations to construct constraints honoring geological

relationships more flexibly. Guo et al. (2016); Guo et al. (2018); Guo et al. (2020); Guo et al. (2021) proposed an explicit-

implicit-integrated 3D geological modelling approach for the geometric fusion of different types of complex geological

structure models; therein, the HRBF-based implicit method was used to model general strata, faults, and folds, and the skinning

method and the free-form surface were used to model local detailed structures. Wang et al. (2018) proposed an implicit

modeling approach to automatically build a 3D model for orebodies by means of spatial HRBF interpolation directly based on

geological borehole data.

However, the above RBF or HRBF interpolants, which use only the on-contact point datasets for each geological interface or

assign an approximate gradient vector for each on-contact point according to its nearest attitude measurements, cannot be

accurately consistent with actual attitude survey data. Moreover, RBF/HRBF-based methods construct implicit field functions

separately for each geological interface and extract the zero value equipotential surfaces to locate the geological interface.

Therefore, it is difficult to maintain topological consistency between geological bodies, let alone to represent their internal

attributes and structural attitudes. Our AdaHRBF interpolation scheme yields an HRBF linear system that is analogous in form



to the previously developed implicit potential field interpolation method based on cokriging of contact increments using parametric isotropic covariance functions.

# 3 Methodology

## 3.1 Modeling Constraints

The geological boundaries and structural attitudes on geological maps and cross-sections are the most common data used for 3D geological modeling. Besides the geological boundaries extracted from boreholes, cross-sections, and geological maps, structural attitude (including strike direction, dip direction, and dip angle) data from geological maps play important roles in characterizing the shape and distribution of geological bodies. As shown in Fig. 1a, stratum S1 is between its bottom surface $f_1$ and top surface $f_2$; a fault interface F divides the 3D space into two sub-domains $D_1$ and $D_2$. We can extract the on-contact boundary points and off-contact attitude points of the strata and fault from the cross-section AA' (Fig. 1b) and geological map (Fig. 1c). The SPF modeling method can jointly reconstruct a 3D geological model using these data extracted from geological maps and cross-sections.

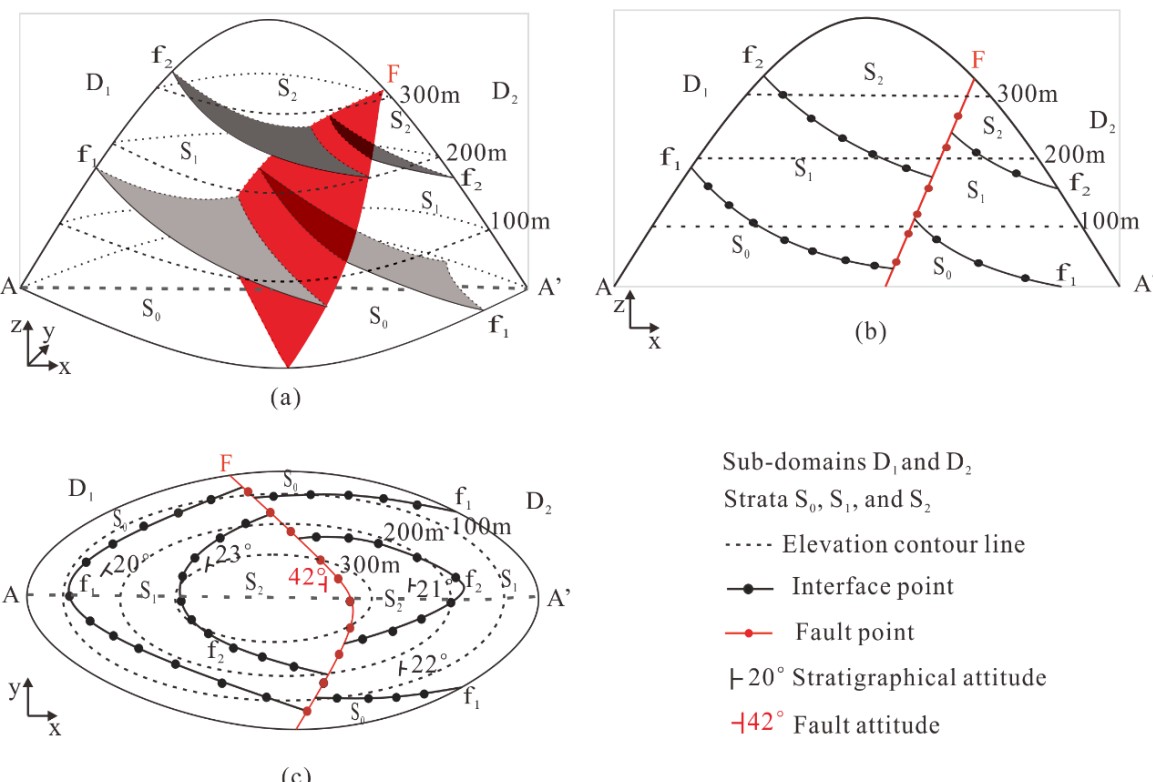

**Figure 1.** Data commonly used in (a) 3D geological modeling extracted from (b) cross-sections and (c) geological maps.





A field in a spatial domain $\mathbb{R}^n$ defines the function $f=f(\boldsymbol{p})$ at a point $\boldsymbol{p} \in \mathbb{R}^n$ in domain $\mathbb{R}^n$, and $f(\boldsymbol{p})$ is also called field function. The SPF defines the 3D space as a scalar function $f(\boldsymbol{p})$ at any point $\boldsymbol{p}$, meanwhile, the stratigraphic interfaces are simulated and expressed as specific equipotential surfaces satisfying $f(\boldsymbol{p})=f_k$ ($i = 1, ..., K$) in the SPF. In practice, this specific function value $f_k$ may correspond to the age of the stratigraphic interface or a relative distance from a reference interface (Mallet, 2004). Therefore, a stratum occupies the space between its bottom surface $f_k$ and top surface $f_{k+1}$, while there are countless disjoint equipotential surfaces in each stratum (Mallet, 2004). A well-known problem is how to interpolate unknown points by a function $f(\cdot)$ using known points of the space $\mathbb{R}^n$. The key problem of SPF modelling is to obtain surfaces that are consistent with known on-contact points on the stratigraphic interfaces and the off-contact attitudes of the strata. The stratigraphic interface points define the distribution of reference equipotential surfaces, while the attitude points define the gradient vectors of the scalar field.

The SPF modeling by the HRBF interpolant satisfies both the on-contact attribute constraint and off-contact attitude constraint. To fit an implicitly defined SPF from known attribute values $\{(\boldsymbol{p}_i, f_i)\}_{i=1}^N \in \mathbb{R}^n \times \mathbb{R}$ and gradients $\{(\boldsymbol{p}_j, \boldsymbol{g}_j)\}_{j=1}^M \in \mathbb{R}^n \times \mathbb{R}^n$ derived from attitude data, we can search for a function $f: \mathbb{R}^n \to \mathbb{R}$ which satisfies both the on-contact constraints $f(\boldsymbol{p}_i) = f_i$ for each $i = 1,...,N$ and the off-contact gradient constraints $\nabla f(\boldsymbol{p}_j) = \boldsymbol{g}_j$ for each $j = 1,..., M$. In particular, $\boldsymbol{p}_i = \begin{bmatrix} p_i^x & p_i^y & p_i^z \end{bmatrix}$ and $\boldsymbol{g}_j = \begin{bmatrix} g_j^x & g_j^y & g_j^z \end{bmatrix}$ in space $\mathbb{R}^3$.

**3.2 HRBF Interpolant**

With the joint constraints of $f(\boldsymbol{p}_i) = f_i$ and $\nabla f(\boldsymbol{p}_j) = \boldsymbol{g}_j$, the optional solution is to obtain equipotential surfaces that are as smooth as possible, that is, to ensure the energy function of SPF, which represents the degree of equipotential surface smoothness and unevenness, as small as possible (Bachman and Narici, 2000). Therefore, the energy function ($E$) of the SPF is defined by the second-order derivative of $f(\boldsymbol{p})$ as:

$$E = \sum_{i=1}^N (f(\boldsymbol{p}_i) - f_i)^2 + \sum_{j=1}^M \left(\frac{\partial f(\boldsymbol{p}_j)}{\partial x} - g_j^x\right)^2 + \left(\frac{\partial f(\boldsymbol{p}_j)}{\partial y} - g_j^y\right)^2 + \left(\frac{\partial f(\boldsymbol{p}_j)}{\partial z} - g_j^z\right)^2$$
$$+ \int_{\mathbb{R}^3} \frac{\partial^2 f(\boldsymbol{p})}{\partial^2 x} + \frac{\partial^2 f(\boldsymbol{p})}{\partial^2 y} + \frac{\partial^2 f(\boldsymbol{p})}{\partial^2 z} + 2\frac{\partial^2 f(\boldsymbol{p})}{\partial x \partial y} + 2\frac{\partial^2 f(\boldsymbol{p})}{\partial y \partial z} + 2\frac{\partial^2 f(\boldsymbol{p})}{\partial z \partial x} \qquad (1)$$

where $\frac{\partial f(\boldsymbol{p}_j)}{\partial x}$, $\frac{\partial f(\boldsymbol{p}_j)}{\partial y}$, and $\frac{\partial f(\boldsymbol{p}_j)}{\partial z}$ are the first-order partial derivatives of implicit function $f(\boldsymbol{p})$ at point $\boldsymbol{p}_j$; $\frac{\partial^2 f(\boldsymbol{p})}{\partial^2 x}$, $\frac{\partial^2 f(\boldsymbol{p})}{\partial^2 y}$,

$\frac{\partial^2 f(\boldsymbol{p})}{\partial^2 z}$, $\frac{\partial^2 f(\boldsymbol{p})}{\partial x \partial y}$, $\frac{\partial^2 f(\boldsymbol{p})}{\partial y \partial z}$, and $\frac{\partial^2 f(\boldsymbol{p})}{\partial z \partial x}$ are the second-order partial derivatives of implicit function $f(\boldsymbol{p})$.

When using the HRBF interpolation method, we usually add a first-order polynomial $C(\boldsymbol{p})$ to ensure the smoothness and continuity of equipotential surfaces. In particular, $C(\mathbf{p}) = c_1 + c_2 \boldsymbol{p}^x + c_3 \boldsymbol{p}^y + c_4 \boldsymbol{p}^z$. The HRBF interpolation function has a





concrete form:

$$f^*(\boldsymbol{p}) = \sum_{i=1}^{N} \alpha_i \varphi(\|\boldsymbol{p} - \boldsymbol{p}_i\|) + \sum_{j=1}^{M} \langle \boldsymbol{\beta}_j, \nabla\varphi(\|\boldsymbol{p} - \boldsymbol{p}_j\|) \rangle + C(\boldsymbol{p}) \qquad (2)$$

$$\nabla f^*(\boldsymbol{p}) = \sum_{i=1}^{N} \alpha_i \nabla\varphi(\|\boldsymbol{p} - \boldsymbol{p}_i\|) + \sum_{j=1}^{M} \nabla^2 \varphi(\|\boldsymbol{p} - \boldsymbol{p}_j\|)\boldsymbol{\beta}_j + \nabla C(\boldsymbol{p}) \qquad (3)$$

where, $\|\boldsymbol{p} - \boldsymbol{p}_i\|$ denotes the Euclidean distance between locations $\boldsymbol{p}$ and $\boldsymbol{p}_i$; $\varphi(r)$ is the radial basis function, herein, for which the cubic function $\varphi(r) = r^3$ was used in this study; $\nabla$ is the Hamiltonian operator; $\nabla^2$ is the Hessian operator, in particular, $\nabla = \begin{bmatrix} \frac{\partial}{\partial x} & \frac{\partial}{\partial y} & \frac{\partial}{\partial z} \end{bmatrix}^T$ and $\nabla^2 = \begin{bmatrix} \frac{\partial^2}{\partial^2 x} & \frac{\partial^2}{\partial x \partial y} & \frac{\partial^2}{\partial x \partial z} \\ \frac{\partial^2}{\partial y \partial x} & \frac{\partial^2}{\partial^2 y} & \frac{\partial^2}{\partial y \partial z} \\ \frac{\partial^2}{\partial z \partial x} & \frac{\partial^2}{\partial z \partial y} & \frac{\partial^2}{\partial^2 z} \end{bmatrix}$; and $\langle \mathbf{a}, \mathbf{b} \rangle$ is the inner product of vectors $\mathbf{a}$ and $\mathbf{b}$. The scalar weight coefficients $\alpha_i \in \mathbb{R}$, vector weight coefficients $\boldsymbol{\beta}_j \in \mathbb{R}^n$, and $\boldsymbol{c} \in \mathbb{R}^{n+1}$ (in particular, $\boldsymbol{\beta}_j = \begin{bmatrix} \beta_j^x & \beta_j^y & \beta_j^z \end{bmatrix}^T$ and $\boldsymbol{c} = [c_1 \quad c_2 \quad c_3 \quad c_4]^T$) are unknown and uniquely determined by the joint constraints $f^*(\boldsymbol{p}_i) = f_i$ for each $i = 1,...,$ N and $\nabla f^*(\boldsymbol{p}_j) = \boldsymbol{g}_j$ for each $j = 1,...,$ M.

The HRBF interpolant defines the implicit function as a sum of chosen basic functions with their linear weights. Furthermore, the type of basic functions (e.g., Gaussian, multi-quadric, and thin plate spline, as shown in Table 1) affects the result of spatial interpolation. We adopt the cubic function as the basis function in this study, i.e., $\varphi(r) = r^3$.

**Table 1.** Common radial basis functions.

| Name of RBF | Definition |
|---|---|
| Gaussian distribution function | $\varphi(r) = exp\,(r^2/\beta^2)$ |
| Multi-quadric function (MQ) | $\varphi(r) = (r^2 + \beta^2)^{1/2}$ |
| Inverse multi-quadric function (IMQ) | $\varphi(r) = (r^2 + \beta^2)^{-1/2}$ |
| Thin plate spline (TPS) | $\varphi(r) = r^{2k-d}\,log\,r$  or  $\varphi(r) = r^{2k-d}$ |
| Cubic function | $\varphi(r) = r^3$ |
| Linear function | $\varphi(r) = r$ |



According to the joint constraints, the weight coefficients $\boldsymbol{\alpha}$, $\boldsymbol{\beta}$, and $\mathbf{c}$ of the interpolant are determined by the following

linear system:

$$\begin{bmatrix} \boldsymbol{\Phi} & \nabla\boldsymbol{\Phi} & \mathbf{C} \\ (\nabla\boldsymbol{\Phi})^{\mathrm{T}} & \nabla^2\boldsymbol{\Phi} & \nabla\mathbf{C} \\ \mathbf{C}^{\mathrm{T}} & (\nabla\mathbf{C})^{\mathrm{T}} & \mathbf{0} \end{bmatrix} \begin{bmatrix} \boldsymbol{\alpha} \\ \boldsymbol{\beta} \\ \mathbf{c} \end{bmatrix} = \begin{bmatrix} \mathbf{f} \\ \mathbf{g} \\ \mathbf{0} \end{bmatrix} \tag{4}$$

where $\boldsymbol{\Phi} = \begin{bmatrix} \varphi_{11} & \varphi_{12} & \cdots & \varphi_{1N} \\ \varphi_{21} & \varphi_{22} & \cdots & \varphi_{2N} \\ \vdots & \vdots & \ddots & \vdots \\ \varphi_{N1} & \varphi_{N2} & \cdots & \varphi_{NN} \end{bmatrix}_{N\times N}$ , whose element $\varphi_{ij} = \varphi(\|\boldsymbol{p}_i - \boldsymbol{p}_j\|)$;

$\nabla\boldsymbol{\Phi} = \begin{bmatrix} \nabla\varphi_{11} & \nabla\varphi_{12} & \cdots & \nabla\varphi_{1M} \\ \nabla\varphi_{21} & \nabla\varphi_{22} & \cdots & \nabla\varphi_{2M} \\ \vdots & \vdots & \ddots & \vdots \\ \nabla\varphi_{N1} & \nabla\varphi_{N2} & \cdots & \nabla\varphi_{NM} \end{bmatrix}_{N\times nM}$ , whose element $\nabla\varphi_{ij} = \nabla\varphi(\|\boldsymbol{p}_i - \boldsymbol{p}_j\|)$;

$\nabla^2\boldsymbol{\Phi} = \begin{bmatrix} \nabla^2\varphi_{11} & \nabla^2\varphi_{12} & \cdots & \nabla^2\varphi_{1M} \\ \nabla^2\varphi_{21} & \nabla^2\varphi_{22} & \cdots & \nabla^2\varphi_{2M} \\ \vdots & \vdots & \ddots & \vdots \\ \nabla^2\varphi_{M1} & \nabla^2\varphi_{M2} & \cdots & \nabla^2\varphi_{MM} \end{bmatrix}_{nM\times nM}$ , whose element $\nabla^2\varphi_{ij} = \nabla^2\varphi(\|\boldsymbol{p}_i - \boldsymbol{p}_j\|)$; $\mathbf{C}=C(\boldsymbol{p})$, in particular, $\mathbf{C} =$

$\begin{bmatrix} 1 & p_1^x & p_1^y & p_1^z \\ 1 & p_2^x & p_2^y & p_2^z \\ \vdots & \vdots & \vdots & \vdots \\ 1 & p_N^x & p_N^y & p_N^z \end{bmatrix}_{N\times(n+1)}$ ;

$\nabla\mathbf{C} = \begin{bmatrix} \mathbf{0} & \nabla p_1^x & \nabla p_1^y & \nabla p_1^z \\ \mathbf{0} & \nabla p_2^x & \nabla p_2^y & \nabla p_2^z \\ \vdots & \vdots & \vdots & \vdots \\ \mathbf{0} & \nabla p_M^x & \nabla p_M^y & \nabla p_M^z \end{bmatrix}_{nM\times(n+1)}$ , whose elements $\nabla p_i^x = [1 \quad 0 \quad 0]^T$, $\nabla p_i^y = [0 \quad 1 \quad 0]^T$, and $\nabla p_i^z = [0 \quad 0 \quad 1]^T$,

respectively.

$\boldsymbol{\alpha} = [\alpha_1 \quad \alpha_2 \quad \cdots \quad \alpha_N]^T$; $\boldsymbol{\beta} = [\boldsymbol{\beta}_1 \quad \boldsymbol{\beta}_2 \quad \ldots \quad \boldsymbol{\beta}_M]^T$;

$\mathbf{f} = [f_1 \quad f_2 \quad \cdots \quad f_N]^T$; and $\boldsymbol{g} = [\boldsymbol{g}_1 \quad \boldsymbol{g}_2 \quad \cdots \quad \boldsymbol{g}_M]^T$.

Once we have the weight coefficients $\alpha_i$, $\boldsymbol{\beta}_j$, and the polynomial coefficients ($c_1$, $c_2$, $c_3$, $c_4$) by solving the above HRBF linear system, we can substitute the weight coefficients and polynomial coefficients into the HRBF equations, then the interpolant function $f(\boldsymbol{p})$ and its gradient function $\nabla f(\boldsymbol{p})$ can be easily obtained.





### 3.3 Adaptive Gradient Constraint

#### 3.3.1 Determination of Gradient Direction

The gradient of the SPF is an important feature of stratum shape, because it indicates the attitude of a stratum. For construction of a scalar field $f(\boldsymbol{p})$, the gradient constraints $\nabla f(\boldsymbol{p}_j) = \boldsymbol{g}_j$ can also be added into modeling process. As shown in Fig. 2, the gradient vector $\boldsymbol{g}$ of SPF and the normal vector $\boldsymbol{n}$ of the stratigraphic interface have the same direction. The gradient vector $\boldsymbol{g}$, the strike vector $\boldsymbol{s}$ and dip vector $\boldsymbol{d}$ of the SPF are orthogonal to each other. The strike $\theta_1$ is the direction of the intersection of the stratigraphic interface and horizontal plane, which is represented by the angle between the strike vector $\boldsymbol{s}$ and the north

direction. The dip $\theta_2$, which is the projected direction of the dip vector $\boldsymbol{d}$ onto the horizontal plane, is represented by the angle between the projected dip direction and the north direction. Strike direction and dip direction are perpendicular to each other. Dip angle $\theta_3$ is the angle between the dip vector and projected dip direction. The three elements form the stratigraphic interface's attitude. The dip angle and strike direction can be obtained through geological observation.

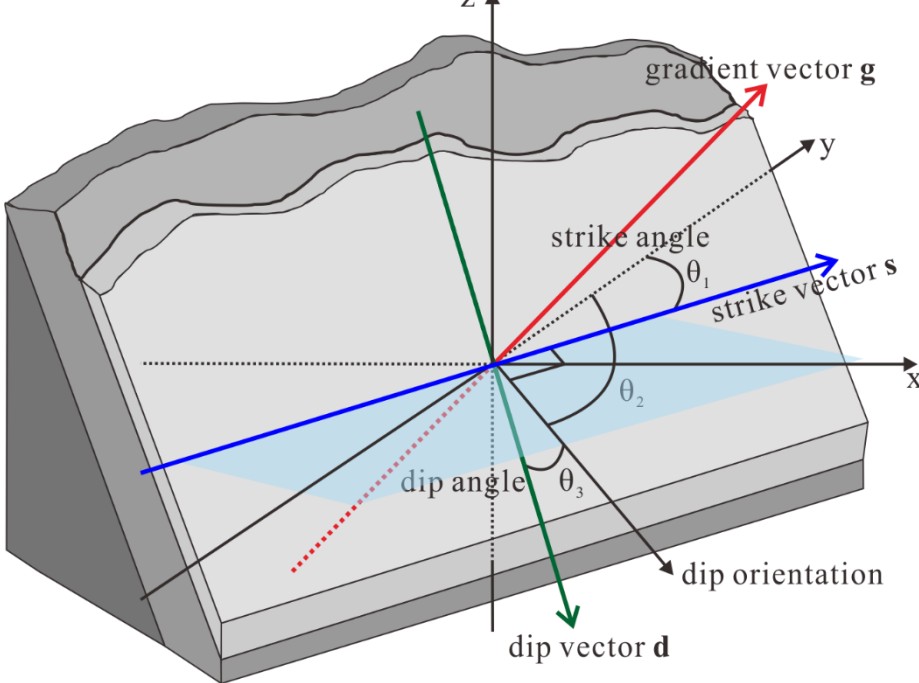


**Figure 2.** The gradient vector **g**, the strike vector **s,** and the dip vector **d**.

The gradient $\boldsymbol{g}$ is a vector with magnitude and direction (which is the same as the normal direction $\boldsymbol{n}$ of the stratigraphic interface). The X-axis, Y-axis, and Z-axis components of the normal direction, $\boldsymbol{n}^x$, $\boldsymbol{n}^y$, and $\boldsymbol{n}^z$, in the 3D Cartesian coordinate system can be derived from the strike, dip and angle of dip of the stratigraphic interface as following:





$$\begin{cases} \boldsymbol{n}^x = \cos\big(radians(\theta_3)\big) * \sin\big(radians(\theta_2)\big) \\ \boldsymbol{n}^y = \cos\big(radians(\theta_3)\big) * \cos\big(radians(\theta_2)\big) \\ \boldsymbol{n}^z = -\sin\big(radians(\theta_3)\big) \end{cases} \qquad (5)$$


### 3.3.2 Optimization of Gradient Magnitude

However, it is difficult to obtain the gradient magnitude through any geological observation. The exact definition of gradient magnitude ($\|\boldsymbol{g}\|$) is the change of an attribute value over unit distance along the gradient direction. The gradient magnitude reflects the rate of change of the scalar field values, which is caused by the difference of stratum thickness at different locations.

As shown in Fig. 3a, a larger gradient magnitude $\boldsymbol{g}_1$ indicates that the stratum becomes thinner, whereas a smaller gradient magnitude $\boldsymbol{g}_3$ indicates that the stratum tends to become thicker. We assume that the gradient magnitude changes gradually everywhere in the scalar field; therefore, every equipotential surface inside of the stratum changes uniformly. However, as shown in Fig. 3b, if we force all the gradient magnitudes to be equal, it may cause the inconsistent SPF changes with neighbors; that is, the trends of some equipotential surfaces inside of the stratum change suddenly compared to other equipotential surfaces.

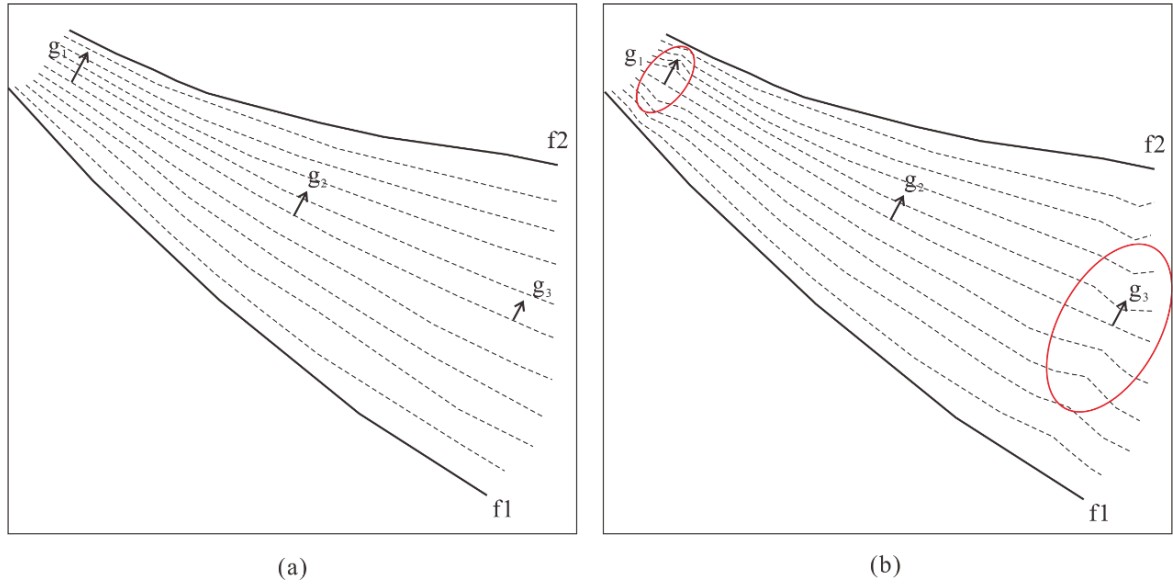


**Figure 3.** Influence of gradient magnitude (indicated by the length of the arrow) on the stratum shape: (a) different gradient magnitudes; (b) the same gradient magnitudes.

It is difficult to determine the exact gradient magnitude through any geological measurement. Because the observation of stratigraphic attitude cannot be used to deduce the gradient magnitude ($l_k$), we added a diagonal matrix $\boldsymbol{\Lambda}$ to the Eq. 4 used

an iterative method to converge on the true gradient magnitude ($l_k$). With $\boldsymbol{\Lambda}=\boldsymbol{0}$, the solution of the Eq. 6 becomes a problem of interpolation by the gradient magnitude $l_k$. With $\boldsymbol{\Lambda}\neq\boldsymbol{0}$, the solution of the Eq. 6 becomes a problem of approximations by the gradient magnitude $l_k$, where diagonal elements of $\boldsymbol{\Lambda}$ represents the degrees of approximations for each gradient





constraint. When $\boldsymbol{\Lambda} \rightarrow 0$, the solution is close to interpolation.

$$\begin{bmatrix} \boldsymbol{\Phi} & \nabla\boldsymbol{\Phi} & \mathbf{C} \\ (\nabla\boldsymbol{\Phi})^{\mathrm{T}} & \nabla^2\boldsymbol{\Phi} + \boldsymbol{\Lambda} & \nabla\mathbf{C} \\ \mathbf{C}^{\mathrm{T}} & (\nabla\mathbf{C})^{\mathrm{T}} & \mathbf{0} \end{bmatrix} \begin{bmatrix} \boldsymbol{\alpha} \\ \boldsymbol{\beta} \\ \mathbf{c} \end{bmatrix} = \begin{bmatrix} \mathbf{f} \\ \mathbf{l} \times \mathbf{n} \\ \mathbf{0} \end{bmatrix} \tag{6}$$

where the diagonal coefficient matrix is given by $\boldsymbol{\Lambda} = \begin{pmatrix} \lambda_1 & 0 & 0 & 0 \\ 0 & \lambda_2 & 0 & 0 \\ \vdots & \vdots & \ddots & \vdots \\ 0 & 0 & 0 & \lambda_{\mathrm{M}} \end{pmatrix}$, in particular, $\boldsymbol{\lambda}_k = \begin{bmatrix} \lambda_k^x & 0 & 0 \\ 0 & \lambda_k^y & 0 \\ 0 & 0 & \lambda_k^z \end{bmatrix}$. Given the

gradient magnitudes $\mathbf{l} = \begin{bmatrix} l_1 & l_2 & \cdots & l_{\mathrm{M}} \end{bmatrix}^{\mathrm{T}}$, then the gradient $\boldsymbol{g}_k = l_k \times \boldsymbol{n}_k$ is the product of $l_k$ and normal vector $\boldsymbol{n}^k$:

$$\sum_{i=1}^{N} \alpha_i \nabla\varphi(\|\boldsymbol{p}_k - \boldsymbol{p}_i\|) + \sum_{j=1}^{M} [\nabla^2 \varphi(\|\boldsymbol{p}_k - \boldsymbol{p}_j\|) + \lambda_k]\boldsymbol{\beta}_j + \nabla C(\boldsymbol{p}_k) = l_k \times \boldsymbol{n}_k \tag{7}$$

We initially set $\boldsymbol{\lambda}_k^{(t=0)}$ to a nonzero constant vector and $l_k^{(t=0)} = 1$. After solving the HRBF system, we can get the function of scalar field $f(\boldsymbol{p})$, then the gradient vector on the attitude observed point $\boldsymbol{p}_k$ is easily obtained according to $\boldsymbol{g}_k = \nabla f(\boldsymbol{p}_k)$.

We record the HRBF coefficients calculated at the $t$-th time as $\alpha_i^t$ and $\boldsymbol{\beta}_j^t$, and record the gradient magnitude at the attitude observed point $\boldsymbol{p}_k$ as $l_k^t$. After solution of the linear system in Equation (6), we estimate the gradient magnitudes $l_k^t$ and the gradient constraint at next iteration step as $\boldsymbol{g}_k^{\mathrm{t}} = l_k^t \times \boldsymbol{n}_k$. Accordingly, we shrink the coefficient $\boldsymbol{\lambda}_k^{t+1}$ to fit more closely to the update gradient constraint. Our idea is that when gradient magnitudes converge, the resulting implicit function interpolates the converged $l_k^t$.

240        In this study, we calculate the increment of $\lambda$ from:

$$\lambda_k^{t+1} = \lambda_k^t - \frac{l_k^t - l_k^{t-1}}{\mathrm{UQ}_{j=1,\ldots M}\left(|l_j^t - l_j^{t-1}|\right)} \tag{8}$$

where UQ() is the upper quartile of differences of all gradient magnitudes. Given the updated $\boldsymbol{\lambda}_k^{t+1}$ and $l_k^t$, we substitute them into the $(t+1)$-th HRBF system (Eq. 6) and solve for the updated coefficient of implicit function. This iterative process continues until the stopping criteria is satisfied.

We use two stopping criteria to finish the iterations. Firstly, for all observed attitude points, if the sum of differences of gradient magnitudes between two consecutive iterations is less than or equal to a small enough threshold ε, we stop the iterations on convergency. Secondly, the number of iterations reaches a given number $N_{\mathrm{iterate}}$, we also obtain the final results of $\alpha_i^t$, $\boldsymbol{\beta}_j^t$, and $l_k^{\mathrm{t}}$.

$$\sum_{k=1}^{M} |l_k^t - l_k^{t-1}| \leq \varepsilon \tag{9}$$

where | | represents the absolute value of a real number and M is the number of observed attitude points. The basic steps of the





iterative calculation of gradient magnitude are given in the pseudo code (Fig. 4).

| | |
|---|---|
| **Input:** | Known attribute value points $\{(\boldsymbol{p}_i, f_i)\}_{i=1}^{N} \in {}^n \times$ ; |
| | Known attitude vector points $\{(\boldsymbol{p}_j, \boldsymbol{n}_j)\}_{j=1}^{M} \in {}^n \times {}^n$. |
| **Output:** | Coefficient $\boldsymbol{\alpha} = [\alpha_1 \quad \alpha_2 \quad \cdots \quad \alpha_N]^T$; |
| | Coefficient $\boldsymbol{\beta} = [\boldsymbol{\beta}_1 \quad \boldsymbol{\beta}_2 \quad ... \quad \boldsymbol{\beta}_M]^T$; |
| | Coefficient $\mathbf{c} = [c_1 \quad c_2 \quad c_3 \quad c_4]^T$. |
| | Gradient magnitude $\boldsymbol{l} = [l_1 \, l_2 \, .. \, l_M]^T$. |
| **Variables:** | Maximum number of iterations: $N_{iterate}$=1000; |
| | No. of current iteration: t=0; |
| | Threshold of termination $\varepsilon = 1e - 5$; |
| | Initial optimization coefficient $\boldsymbol{\lambda}_k^{(t=0)}$; |
| | Initial gradient magnitude $l_k^{(t=0)}$; |
| | Absolute error of the gradient magnitudes between two adjacent iterations $r^t$. |
| **Steps:** | |
| 1. | while (t < $N_{iterate}$ and $r^t > \varepsilon$) do |
| 2. | Add disturbance $\boldsymbol{\lambda}_k^t$ to calculate the coefficients $\boldsymbol{\alpha}$, $\boldsymbol{\beta}$ and $\mathbf{c}$. |
| 3. | t = t + 1. |
| 4. | Calculate known points $\boldsymbol{g}_k$ by $\boldsymbol{\alpha}$, $\boldsymbol{\beta}$ and $\mathbf{c}$. |
| 5. | for (1 to M) do |
| 6. | Calculate $l_k^t$ and $r^t$ at each known attitude point. |
| 7. | end for |
| 8. | Get the upper quartile of $r^t$. |
| 9. | for (1 to M) do |
| 10. | Calculate $\boldsymbol{\lambda}_k^t$ at each known attitude point. |
| 11. | end for |
| 12. | end while |
| 13. | return $\boldsymbol{\alpha}$, $\boldsymbol{\beta}$, $\mathbf{c}$, and $\boldsymbol{l}$. |

**Figure 4**. Pseudo code of iterative algorithm for optimizing gradient magnitude.

## 4 Verification Experiments

Two experimental fields in 2D space, with gradient changing in direction or magnitude, were designed to verify the AdaHRBF method. The experimental results show that the different gradient magnitude settings apparently affect the modeled fields, moreover, the AdaHRBF method is effective to iteratively obtain the true gradient magnitude of the fields. We modeled an





analytic field of $f_1(\boldsymbol{p}) = ((\boldsymbol{p}^x - 300)^2 + (\boldsymbol{p}^y)^2)^{\frac{3}{2}}$ with the changing gradient direction and magnitude as show in Fig. 5a.

Then we sampled attribute and attitude points from the analytic field with different locations as shown in Fig. 5b. Hence, we

can retrieve the coefficients $\alpha_i$ and $\boldsymbol{\beta}_j$ of the HRBF formula and the polynomial coefficients, respectively. We compared two different experimental settings: (1) Assuming that gradient is a unit vector and each gradient magnitude is 1, we used the HRBF interpolant to reconstruct the field as shown in Fig. 5c. Although the field values at the sampling points are equal to the given attribute values, the retrieved field values change irregularly, thus we obtained a large number of exceptional values in the reconstructed field. (2) The true gradient magnitude was obtained via the iterative AdaHRBF method introduced above. In

this condition, we more accurately restored the field (as shown in Fig. 5d) and also got the optimized gradient magnitude after the iterations, which was close to the true value.

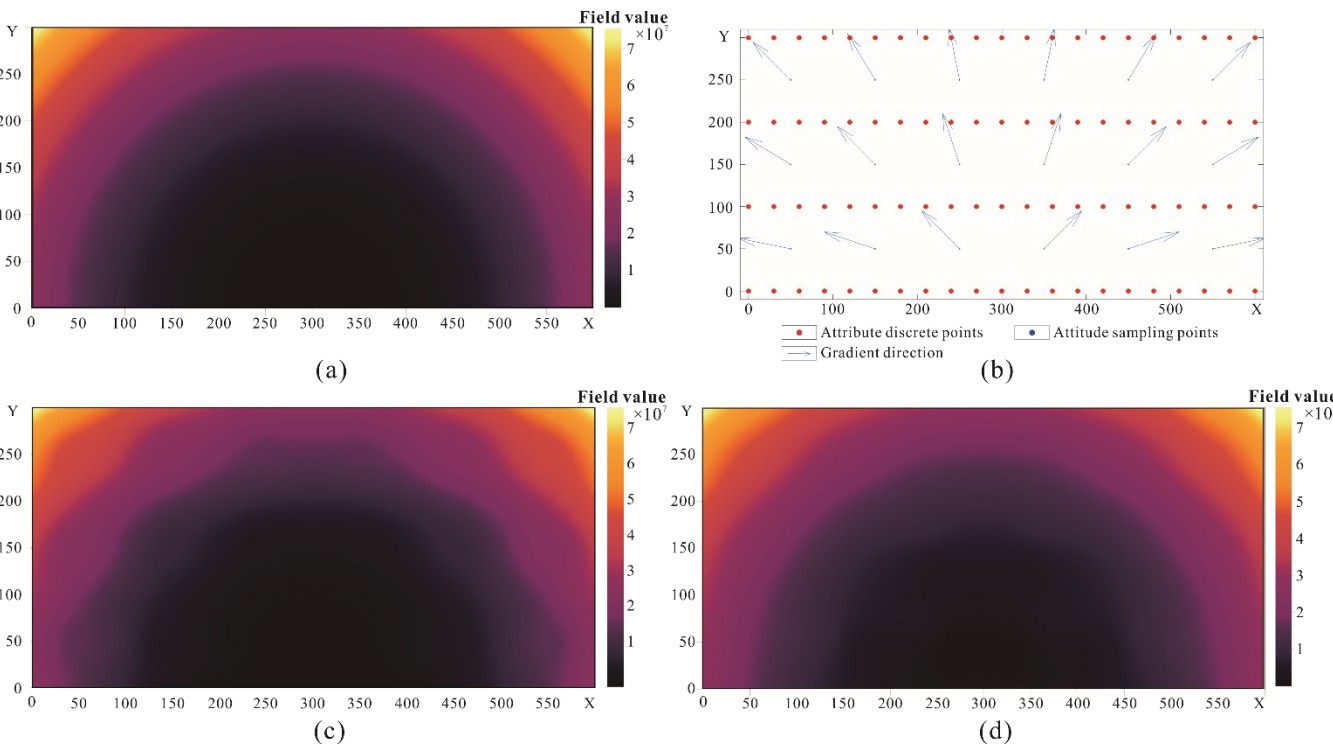

**Figure 5.** Experimental field 1: (a) Original potential field; (b) distribution of attribute and attitude points; (c) field reconstructed when each gradient magnitude was set to a constant value of 1; and (d) field reconstructed when the gradient magnitude was obtained iteratively.


We also modeled a potential field of $f_2(\boldsymbol{p}) = (\boldsymbol{p}^y)^3$ with the changing gradient magnitude as show in Fig. 6a. It is known that each direction of gradient points is the positive Y-axis direction. We sampled attribute points and attitude points as shown in Fig. 6b. We also compared two different experimental conditions: (1) Assuming that each fixed gradient magnitude is 1, we used the HRBF interpolant to reconstruct the field as shown in Fig. 6c. (2) The true gradient magnitude was obtained via the





iterative AdaHRBF method. In this condition, we more accurately restored the potential field (as shown in Fig. 6d) and also

got the true gradient magnitude after the iterations.

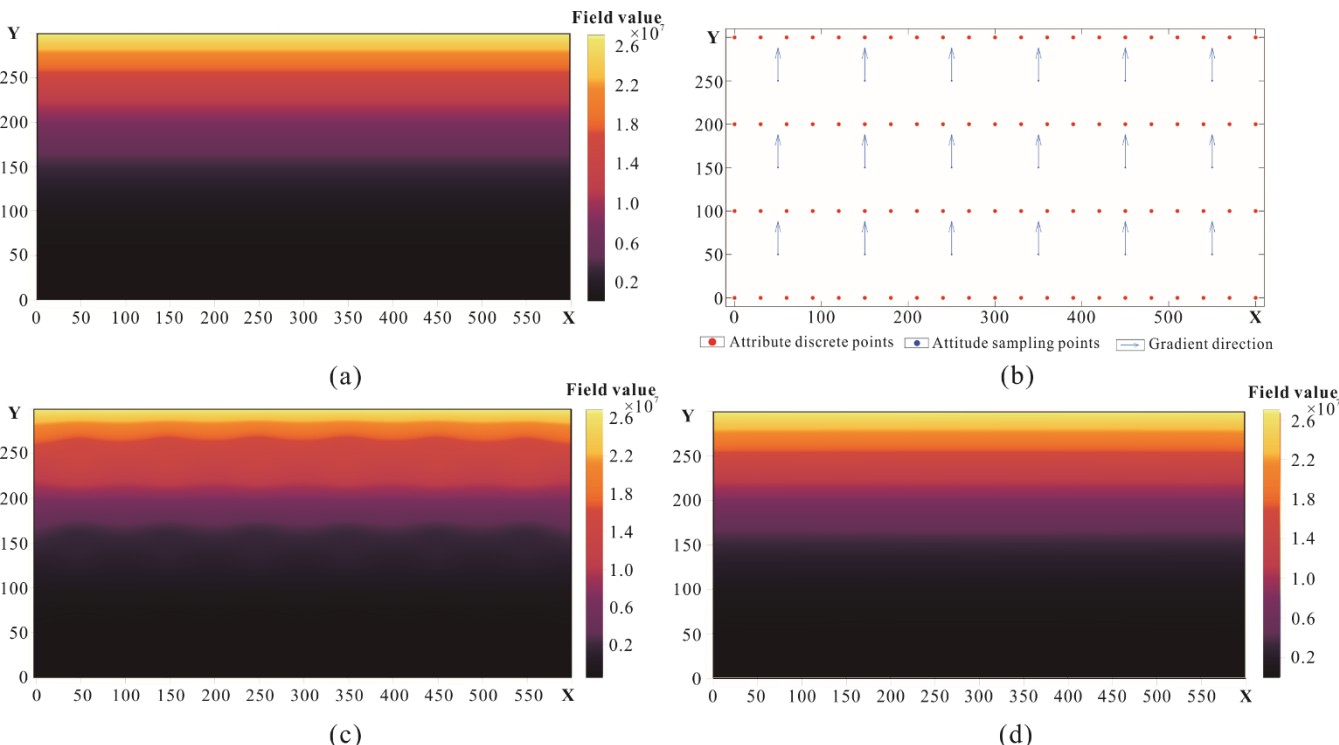

**Figure 6.** Experimental field 2: (a) Original potential field; (b) distribution of attribute and attitude points; (c) field reconstructed when each gradient magnitude was set to a constant value of 1; and (d) field reconstructed when the gradient magnitude was obtained iteratively.


We overlaid above-mentioned two fields to generate a new potential field of $f_3(\boldsymbol{p}) = f_1(\boldsymbol{p}) + f_2(\boldsymbol{p})$ as show in Fig. 7a, and

the sampled attribute points and attitude points are shown in Fig. 7b. We also compared two different experimental conditions:

(1) Assuming that each fixed gradient magnitude is 1, we used the HRBF interpolant to reconstruct the field as shown in Fig.

7c. (2) The true gradient magnitude was obtained via the iterative AdaHRBF method, and we more accurately restored the

potential field (as shown in Fig. 7d).



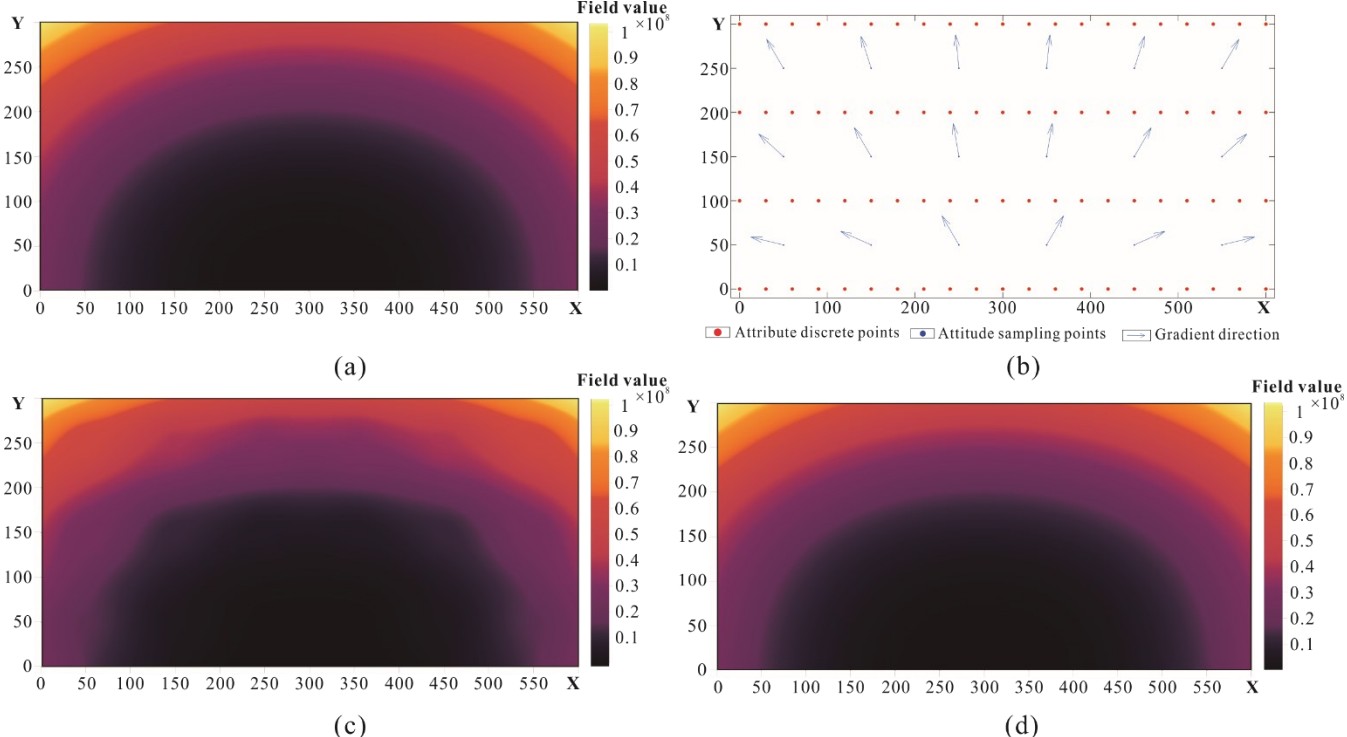

**Figure 7.** Experimental field 3: (a) Original potential field; (b) distribution of attribute and attitude points; (c) field reconstructed when each gradient magnitude was set to a constant value of 1; and (d) field reconstructed when the gradient magnitude was obtained iteratively.

## 5 Case Study

### 5.1 Study Area and Dataset

The study area is located in the Lingnian-Ningping manganese ore zone, in Debao County, southwestern Guangxi Zhuang Autonomous Region, China (Fig. 8). The study area mainly consists of strata from the late Paleozoic to the late Triassic-Pliocene ($T_3$-$N_2$). The middle Permian ($P_2$) strata are in para-unconformity contact with early Triassic ($T_1$) strata; the middle Triassic ($T_2$) strata are in angular unconformity contact with Quaternary. There is a left strike-slip inverse fault, the Nacha Fault, in the middle of the study area. It dips to the southeast, with a NE strike direction of 45°, a dip angle of about 70°, and a total length of about 12 km, extending outside the study area. The footwall slid to the west relative to the hanging wall, and the slip distance is about 600 m. There are two synclines (I and III) and an anticline (II) in the study area. Syncline III is located in the middle of the study area with a high symmetry. The axis of syncline III strikes nearly northeast and its south limb is cut by the Nacha Fault. Anticline II is located in the northwest of the study area with a good symmetry, the fold axis striking about 30° northeast.



**Figure 8.** Geological map of the study area.

Faults, unconformable strata, and intrusive rocks all cause discontinuities in a SPF (Calcagno et al., 2008). We used the fault

surface samplings to interpolate the potential field of the Nacha Fault (Fig. 9a). We extracted the zero equipotential surface of

the fault potential field to reconstruct the surface model of the Nacha Fault which divides the study area into two sub-domains

(Fig. 9b). In each sub-domain, the coefficients of the HRBF linear system were separately solved according to the joint

samplings of the SPF and its gradient.



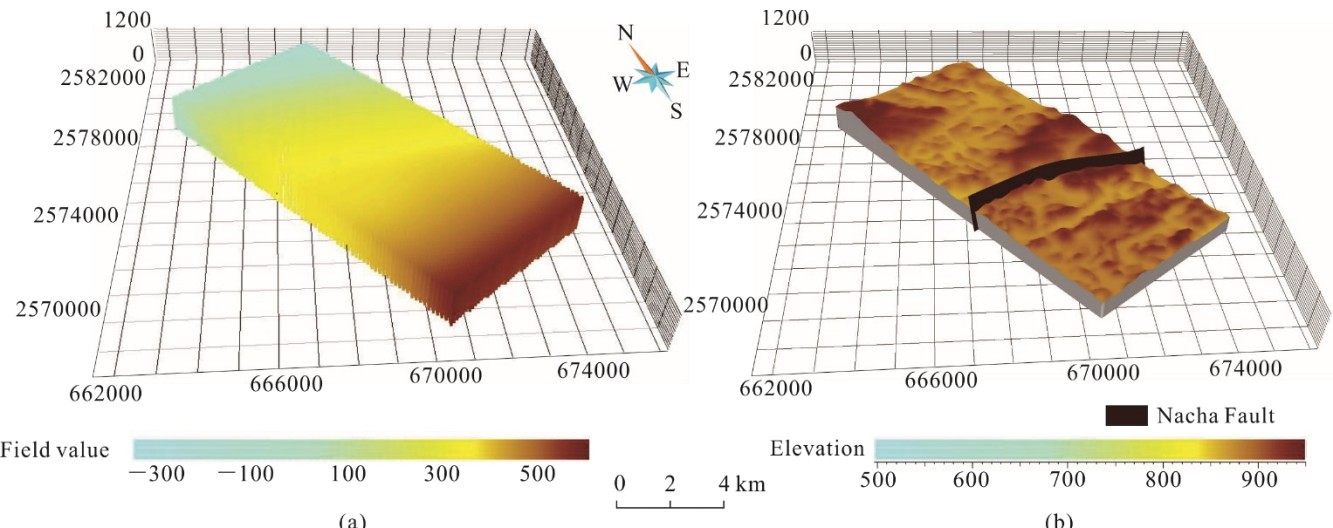

**Figure 9.** Model of Nacha Fault: (a) potential field; and (b) surface model.

According to the comprehensive stratigraphic column, the burial depth of each stratigraphic interface relative to the top surface of the Quaternary was used as the attribute value of the SPF (Fig. 10) for implicit function interpolation. The SPF defines the 3D space as a scalar function $f(p)$ at any point $p$, where $f$ is defined as the relative burial depth in this study. Burial depth decreases as geological time progresses; therefore, earlier deposited strata are assigned a relatively larger burial depth, while later deposited strata are assigned a relatively smaller burial depth. Each point inside of a stratum has its own burial depth relative to the top surface of the Quaternary, therefore, the depth values in the field decrease gradually from bottom to top in strata. In this context, the SPF is fitted by a scalar function of the relative burial depth. When the relative burial depth is used as the attribute value of the SPF, we can set the initial gradient magnitude $\|g\| \cong 1$ if the strata underwent heterogenous deformation. However, if we use geological age as the attribute value of the SPF, $\|g\|$ can no longer be initially assumed to be 1 because the stratigraphic age and distance along the gradient direction are from different measured variables.



| Systems | Series | Stages | Groups | Formations | Members | Mark | Lithologic Colum | Relative Depth (meter) | Lithologic Description |
|---|---|---|---|---|---|---|---|---|---|
| Quaternary System | | | | | | Q | | 0 / −20 | Gravel, sand and clay layers |
| Triassic | Middle | | | Baifeng | Upper | $T_2b^2$ | | −1898 | The upper part is gray-green shale, calcareous mudstone, and siltstone. The lower part is feldspar quartz sandstone. |
| | | | | | Lower | $T_2b^1$ | | −3734 | The upper part is gray-green feldspar quartz sandstone and mudstone, calcareous mudstone interbedded. The middle part is feldspar quartz sandstone intercalated with shale and limestone lens.The lower part is shale. |
| | Lower | | Luolou | Beisi | | $T_1b$ | | −4872 | The upper part is light gray thick layered limestone and dolomitic limestone. The lower part is composed of medium-thick layered dolomite with acid tuff. The Beisi Formation is an important manganese-bearing horizon. It contains 13 manganese ore layers, which are divided into three layers: upper, middle and lower. The ore is medium and low grade manganese carbonate ore and manganese oxide ore. |
| | | | | | | | | | Gray to dark gray, thin to middle layered argillaceous banded limestone, oolitic limestone, intercalated dolomite and acid vitreous limestone. |
| | | | | Majiao-lingian | | $T_1m$ | | −6772 | Flintstone, the bottom exists coal, bauxite or iron aluminum rock. |
| | | | | | | | | −7233 | Thick gray limestone, flint limestone, dark gray dolomite, dolomitic limestone intercalated with limestone |
| Permian | Upper | | | | | $P_2$ | | −7609 | Light gray to gray black medium-thick layered limestone with dolomite, gray to dark gray medium-thin layered flint limestone. |
| | Lower | | | Maokou | | $P_1m$ | | −8214 | Light gray thick limestone intercalated with dolomite. Top and bottom are dolomite intercalated limestone |
| | | | | Chihsia | | $P_1q$ | | −8429 | |
| Carboniferous | Upper | | | | | $C_3$ | | −9145 | The upper part of the Lower Carboniferous is gray thick limestone with dolomite, dolomitic limestone and flint limestone. The Huanglong Formation is light gray, with bioclastic limestone intercalated with dolomite. |
| | Middle | | | | | $C_2$ | | −9674 | |
| | Lower | | | | | $C_1$ | | −10602 | The upper part of the Lower Calcareous System is gray thick limestone and flintbearing limestone. Siliceous rock interbedded with siliceous mudstone in the lower part. The Datang Stage is an important manganese-bearing horizon with multiple layers of manganese ore. There is unstable gravel limestone at the bottom of intercalated siliceous limestone. |
| Devonian | Upper | | | | | $D_3$ | | −11142 | The upper Devonian system is limestone. The lower lentil-like limestone is intercalated with argillaceous limestone. The important manganese-bearing layer of Wuzhishan Formation is divided into three lithological sections from bottom to top according to lithology and its relationship with manganese ore. The first section: purple gray to blue gray siliceous limestone. The second section: light gray to gray thin lime siliceous rock. At the bottom is the third section: dark gray to gray black thin lime siliceous rock. |
| | Middle | Donggang-liangian | | | | $D_2d$ | | −11717 | The upper part is light gray to dark gray, medium-thick layered to massive limestone, flint rock and siliceous rock. The lower part is light gray to gray black dolomite with dolomitic limestone. |
| | Lower | Yujian-gian | | | | $D_1y$ | | −12020 | Gray-green, yellow-green siltstone, argillaceous siltstone, shale, phosphorusbearing sandstone and phosphorus-bearing silty mudstone in the lower part. |
| | | | | Nakaoling | | $D_1n$ | | | Gray-green, yellow-red sandy shale, shale, argillaceous siltstone. |

Scale 1:12500

**Figure 10.** Comprehensive stratigraphic column of the study area.

Based on the geological map and DEM of the study area, we produced a series of cross-sections (Fig. 11). However, the cross-sections were presented in 2D form. According to the necessary geographic projection parameters and scale, therefore, we derived the mapping relationship between 2D and 3D. Finally, we extracted the geological boundary points with 3D coordinates from 2D cross-sections.



**Figure 11.** Geological cross-sections of study area.

The attribute points and attitude points of each stratigraphic interface and fault plane extracted from the geological map and cross-sections were used as the original dataset for 3D SPF modeling. The 3D points of stratigraphic interfaces extracted from the geological map and cross-sections were regarded as samplings of the SPF. The gradient vectors which are transformed from the off-contact stratigraphic attitude points were regarded as the samplings of the gradient of SPF.

## 5.2 Optimizing Gradient Magnitude

There are 1410 known on-contact attribute points and 34 off-contact attitude points scattered throughout the study area (Fig. 12a). The known attitude sampling points are scattered on the south limb of fold I, the north and south limbs of fold II, and the north and south limbs of fold III. There are 17 attitude sampling points in the north side of the Nacha fault and 17 attitude sampling points on the south side. The distribution of the dip directions and dip angles is shown in Fig.12b.




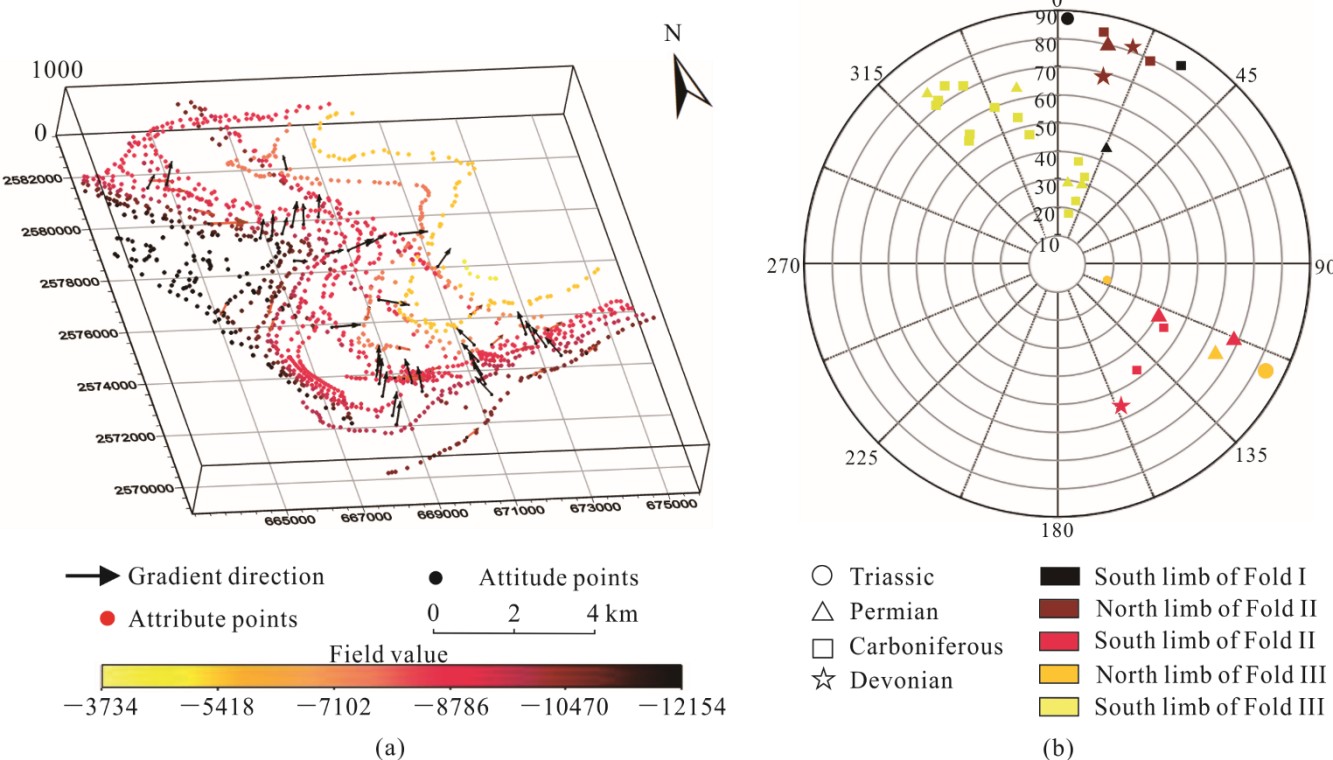

**Figure 12.** Scattered attribute points and attitude points of strata: (a) known attribute points and attitude points of strata; and (b) distribution of the dip directions and dip angles of the attitude points, in which the symbols represent different strata, and the colors represent different limbs of folds.

First, we set the initial gradient magnitude to 1.0, and calculated the X, Y and Z axis components of the gradient vector field according to the dip direction and angle of the attitude points. We constructed HRBF solution matrices on the north and south side of the Nacha Fault, respectively. Then, we iterated to converge toward the true gradient magnitudes by adding an optimization term to the HRBF linear system. The termination conditions were met after 200 iterations in the north sub-domain and 300 iterations in the south sub-domain. The gradient magnitudes became stable, and finally the true magnitudes of gradient were obtained. The changes of gradient magnitude is shown in Fig. 13.



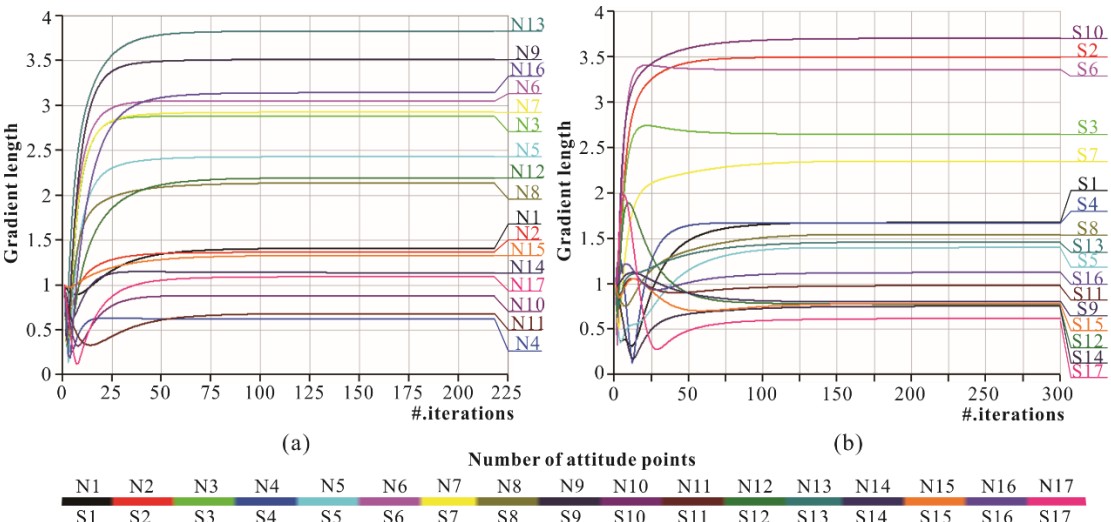

**Figure 13.** Changes of optimization coefficient $\lambda$ and gradient magnitude: (a) gradient magnitudes for all attitude points in the north sub-domain; (b) gradient magnitudes for all attitude points in the south sub-domain. The corresponding number of attitude point can be found in Figure 7.

On a specific grid resolution, we modeled the scalar field of gradient magnitude before and after optimization for each attitude point (Fig. 14). Along the north side of the Nacha Fault in Fig. 14a, the gradient magnitudes obtained by interpolation in area B exceed the maximum values. Compared with the scalar field of gradient magnitude before optimization, the scalar field of gradient magnitude after optimization (Fig. 14b) more smoothly represents changes in the strata. The Carboniferous strata have the largest true gradient magnitude, while he true gradient magnitudes of the Devonian strata are smallest. Furthermore, we cut four cross-sections of the gradient magnitude scalar field, as shown in Fig. 15.

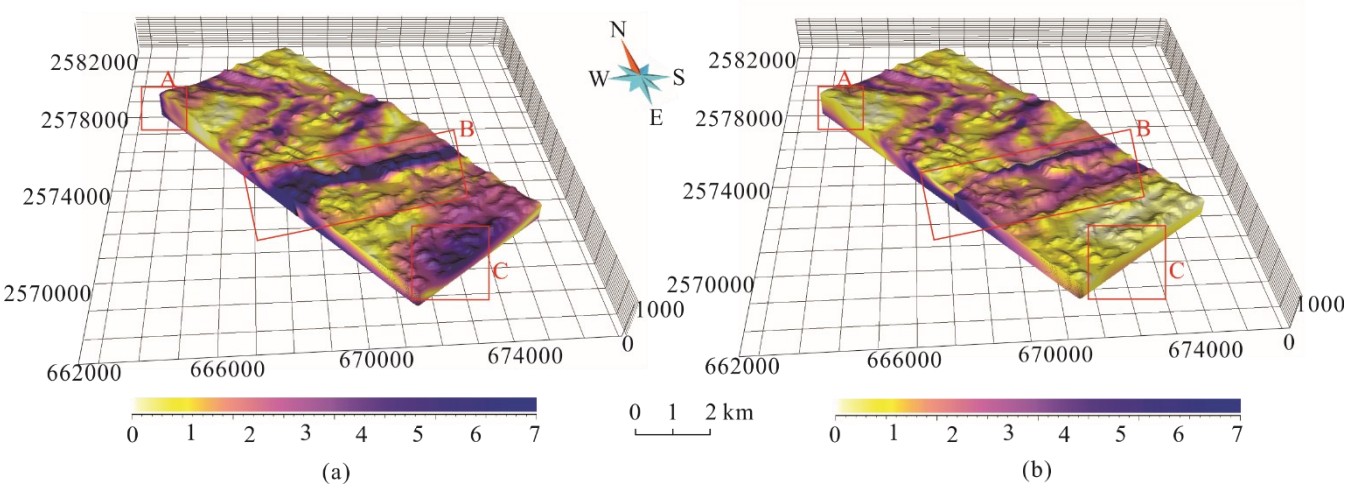

**Figure 14.** Scalar field of (a) gradient magnitude assigning an initial fixed gradient magnitude of 1 for each attitude point; and (b) gradient magnitude after optimization.





**Figure 15.** Cross-sections of the gradient magnitude field: (a) assigning an initial fixed gradient magnitude of 1 for each attitude point; and

(b) after optimization.




### 5.3 Stratigraphic Potential Field (SPF)

After the optimized gradient magnitude for each attitude point was obtained, all scatted attribute points and attitude points were finally substituted into HRBF linear system to respectively solve the HRBF coefficients ($\alpha_i$, $\boldsymbol{\beta}_j$) and the polynomial coefficients ($c_1, c_2, c_3, c_4$) for each side of the Nacha Fault. On a specific grid resolution, we generated the regular discrete

grids as interpolated points in 3D space. Then the points above the digital elevation model (DEM) were removed from the interpolated points. Finally, we reconstruct the SPFs in 3D space before and after optimization of the gradient magnitude according to the respective HRBF interpolant of each sub-domain (Fig. 15). In this study, the SPF represents the relative burial depth in 3D space. The larger field value represents earlier deposited strata with larger relative burial depth, and vice versa. The same stratigraphic interfaces in different sub-domains share the same field value. The field values change abruptly at the

Nacha Fault because the conformable strata were cut by the fault plane.

The SPFs are both constrained so that the interpolated SPFs values at the attribute points are equal to the initial relative burial depths, but the SPFs values may abruptly change or produce outliers at some locations. Obviously, the SPF values change nonuniformly with gradient magnitude before optimization, which caused the SPF values that originally belonged to the Carboniferous strata to be interpolated as those of other strata and sequentially resulted in incorrectly extraction of the

stratigraphic interfaces. The abnormal SPF values (areas A, B and C in Fig. 16a), are not continuously distributed along stratigraphic interfaces but appear at irregular intervals. This abnormal stratigraphic potential field causes separated, discontinuous, and dispersed stratigraphic interfaces to be extracted through equipotential surface tracking. However, reconstructing the SPF through optimization of gradient magnitude for each attitude point (Fig. 16b) avoids the generation of either abnormal field values or of the wrong equipotential surfaces. This geologically plausible SPF can be appropriately

constrained by the known gradient direction and the optimized gradient magnitude at the attitude sampling points.

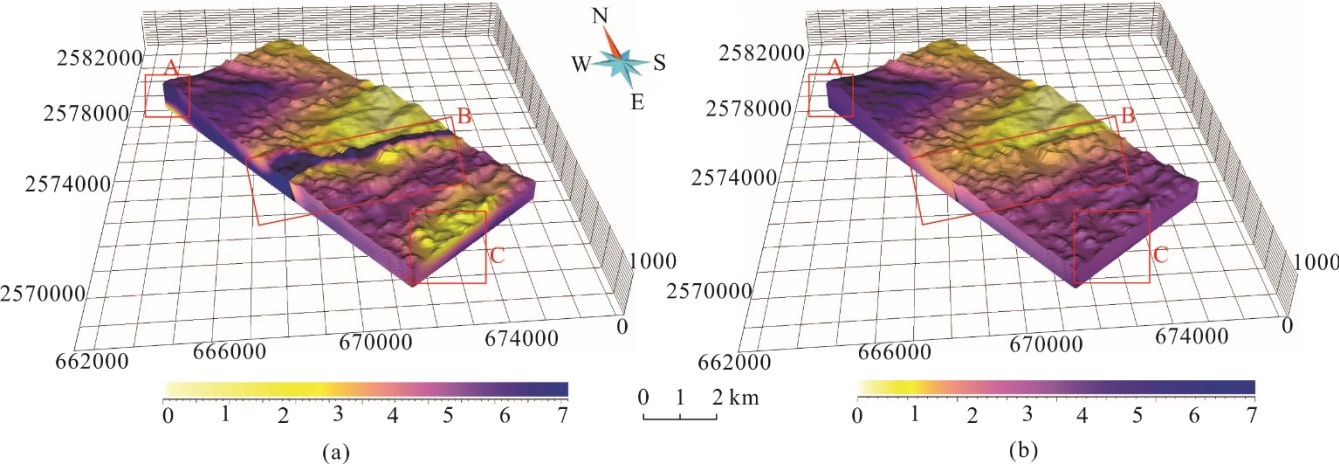

**Figure 16.** Stratigraphic potential field (a) before and (b) after optimization of gradient magnitude.



We cut the SPF along four section lines, and the SPF value also changes more uniformly from older to younger strata after

gradient magnitude optimization than using a fixed gradient magnitude of 1, as shown in Fig. 17.

**Figure 17.** Cross-sections of the stratigraphic potential field (a) before and (b) after optimization of gradient magnitude.



## 5.4 Three-Dimensional Models of Strata

Once the field was interpolated in 3D space, the specific equipotential surfaces were extracted from the implicit volumetric
function as stratigraphic interfaces within each main structure bounded sub-domain. We used the marching cube method to
extract the equipotential surfaces with a specific relative burial depth from the stratigraphic interfaces by connecting all the
points with the same field value in the stratigraphic potential field (Fig. 18). The 3D surface model extracted from the potential
field shows that the geometrical shape of each equipotential (iso-depth) surface is smooth, and the topology is consistent. The
interface model on both sides of the Nacha Fault restores the location of the fault in the south limb of syncline III.



**Figure 18.** Three-dimensional model of the bottom surfaces of strata.

Sequentially, according to the range of relative burial depth of stratigraphic top and bottom, two stratigraphic solid models
were reconstructed from these equipotential surfaces before and after optimization of gradient magnitude for each attitude
point, respectively, combined with sub-domain boundaries and DEM (Fig. 19). Many abnormal potential field values and





additional unreasonable geological bodies were extracted from the model before optimization, especially in areas A and C as shown in Fig. 19a. These abnormal potential field values lead to the occurrences of additional strata fragments that do not conform to the rule of sediments. Therefore, the HRBF interpolation with the initial fixed gradient magnitude of 1 roughly reflects stratigraphic on-contact information and captures the structure of syncline I in the north. However, several details are

different from the stratigraphic structure on the geological map. Where the Nacha Fault passes through syncline III, the strata on the south side of the fault plane should correspond to the same strata on the north side. However, the Devonian strata corresponded to the Permian strata in area B as shown in Fig. 18a, which is inconsistent with the geological structure. The geological model extracted using the optimized gradient magnitude for each attitude point is shown in Fig. 19b. Overall, the obtained geometries follow more closely the shape of the folds and stratigraphic on-contact lines. From north to south in the

study area, anticline II and syncline III were successfully modeled with the Nacha Fault correctly represented as an inverse fault that cuts syncline III. On both sides of Nacha Fault, the sequence of the strata is the same, and the model exhibits traces of the fault plane passing through the stratigraphic surfaces.





(a)

(b)

**Figure 19.** Three-dimensional stratigraphic volume model (a) before optimizing gradient magnitude, and (b) after optimizing gradient magnitude.





Four cross-sections through the solid models (see the geological map for cross-section lines) were cut, and the cross-sections of the solid model are more consistent with the original structural relationships on the geological map after gradient magnitude optimization than using a fixed gradient magnitude of 1, as shown in Fig. 20.

**Figure 20.** Cross-sections of the solid models (a) before and (b) after optimizing gradient magnitude.





The highest stratum and section coincidence percentages on cross-sections are 74.50% ($T_2$) and 78.03% (Section 16) before optimization, respectively, as shown in Table 2. However, the highest stratum and section coincidence percentages on cross-sections are 98.99% ($D_1$) and 98.01% (Sections 13 and 15) using the optimized gradient magnitude for each attitude point, respectively, as shown in Table 3. The total coincidence percentage on cross-sections increases from 67.03% to 98.27% after

optimizing gradient magnitude.

**Table 2.** Coincidence percentages on cross-sections with an initial fixed gradient magnitude of 1 for each attitude point.

| Stratum | Section 13 | Section 14 | Section 15 | Section 16 | **Total** |
|---|---|---|---|---|---|
| $T_2$ | \ | \ | 78.14% | 73.27% | 74.50% |
| $T_1$ | \ | 78.35% | 70.74% | 77.54% | 74.48% |
| $P_1$ | 13.66% | 47.32% | 68.13% | 77.84% | 60.90% |
| $C_3$ | 15.01% | 57.26% | 76.80% | 78.74% | 64.26% |
| $C_2$ | 13.53% | 53.57% | 74.13% | 91.83% | 63.15% |
| $C_1$ | 18.84% | 80.65% | 81.10% | 76.12% | 63.50% |
| $D_3$ | 75.62% | 53.27% | 61.53% | 77.99% | 67.08% |
| $D_2d$ | 12.92% | 66.21% | \ | \ | 37.91% |
| $D_1$ | 82.11% | 66.58% | \ | \ | 74.47% |
| **Total** | 57.84% | 60.58% | 72.13% | 78.03% | 67.03% |

**Table 3.** Coincidence percentages on cross-sections after optimizing gradient magnitude.

| Stratum | Section 13 | Section 14 | Section 15 | Section 16 | **Total** |
|---|---|---|---|---|---|
| $T_2$ | \ | \ | 99.40% | 96.26% | 97.06% |
| $T_1$ | \ | 98.07% | 99.30% | 97.04% | 98.14% |
| $P_1$ | 99.32% | 98.20% | 97.89% | 95.18% | 97.12% |
| $C_3$ | 97.17% | 90.66% | 97.03% | 92.47% | 94.00% |
| $C_2$ | 94.82% | 95.28% | 95.53% | 94.55% | 95.08% |
| $C_1$ | 96.30% | 98.47% | 97.40% | 96.20% | 97.22% |
| $D_3$ | 97.68% | 98.58% | 99.12% | 98.77% | 98.41% |
| $D_2d$ | 96.65% | 91.17% | \ | \ | 94.08% |
| $D_1$ | 99.41% | 98.55% | \ | \ | 98.99% |
| **Total** | 98.01% | 97.22% | 98.01% | 95.90% | 97.27% |



## 5.5 Stratigraphic Thickness Index (STI)

In order to represent the variability of stratigraphic thickness, we introduce a definition of stratigraphic thickness index (STI) at any location $p$ in a stratum:

$$STI(p) = \frac{\left| f_{top}(p) - f_{bottom}(p) \right|}{l(p)} \qquad (10)$$

where $f_{top}(p)$ and $f_{bottom}(p)$ are the potential field values of the top and bottom surfaces, respectively, of a stratum where the point $p$ is located, and $l(p)$ is the gradient magnitude obtained by the previously described iterations at the location $p$. The normalized STI represents the true thickness of the stratum passing through the location $p$; therefore, we can analyze stratigraphic thinning and thickening effects by comparing STI at different locations in the stratum. For each stratum in the study area, the STI of each stratum may be different everywhere.

The STI, which is the ratio of the difference between the potential field values of a stratum's top and bottom surfaces to the gradient magnitude at each point, is a normalized indicator to represent the thickness variation of strata. We restored the STI scalar field using the optimized gradient magnitude and found that the STI values depend on the strata in the study area (Fig. 21). The STI values of the strata of Baifeng Formation, Maokou Formation and Yujiang Stage on the north side of Nacha Fault and the strata of Lower Series of Carboniferous and Upper Series of Devonian on the south side of Nacha Fault are larger than others and distributed in a patchy form. The gradient magnitude of each attitude point before optimization is a fixed value which may not be the true gradient magnitude. If the gradient magnitudes of all attitude points located in different strata were equal, the STI values of points in each stratum would tend to be the same value, which would cause the SPF to vary nonuniformly.

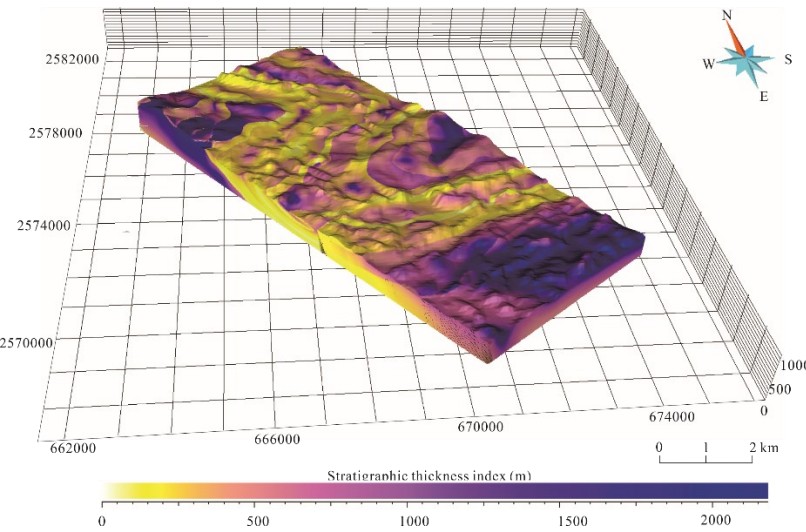

**Figure 21.** Scalar field of stratigraphic thickness index (STI).





We overlaid STI on the geological map, as shown in Fig. 22. Although the difference between the SPF values of a stratum's top and bottom surfaces is a constant, the gradient magnitude varies at different points in a stratum, so the STI also changes laterally. On each side of the Nacha Fault, the STI values of Devonian strata are much greater than other strata; in contrast, the STI values of Carboniferous strata are smaller. This result is consistent with our measured stratigraphic thickness in the comprehensive stratigraphic column of the study area.

**Figure 22.** Values of stratigraphic thickness index overlaid on the geological map.



We cut the STI scalar field along four section lines, as shown in Fig. 23. The STI at the core of syncline III gradually decreases from the northeast to the southwest, which reflects the southwest plunge of this syncline. At the core of anticline II, the STI is lower in the northeast, but is higher in the southwest where the thicker Devonian strata occur, which reflects the northeast

plunge of this anticline.

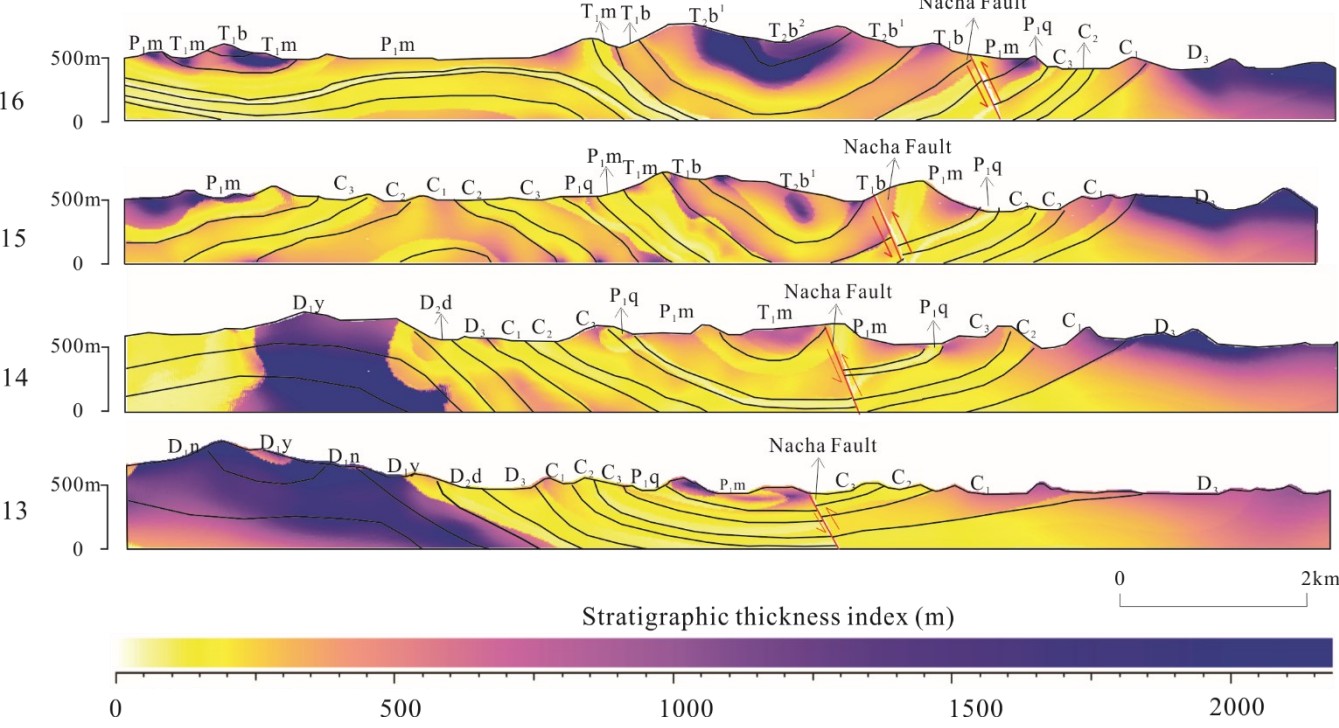

**Figure 23.** Cross-sections of the stratigraphic thickness index (STI) field.

## 6 Discussions

Due to their robustness and stability, RBF and HRBF interpolants are a good choice for modeling when geological sampling

data is relatively sparse and uneven (Cowan et al., 2003; Guo et al., 2016). However, existing RBF and HRBF interpolants implicitly reconstruct a single geological interface and extract it as the zero-value equipotential surface. Moreover, existing RBF and HRBF interpolants need several independent scalar fields to simulate geological interfaces, which makes it difficult to ensure topological consistency among different geological interfaces. The AdaHRBF proposed in this study can create only one scalar field and extract multiple continuous stratigraphic interfaces for a set of conformable strata.

The AdaHRBF proposed in this study improves the use of altitude data in SPF modeling by optimization of gradient magnitudes. In additional to use of attitude information as the gradient directions of SPFs, we use the gradient magnitude as a new constraint to control the rate of change of SPF values. The gradient of a SPF is a vector with certain direction and magnitude, in which



the gradient magnitude provides constraints on the thickness of deformed strata. Therefore, it is extremely important to construct HRBF linear systems with accurate gradient magnitudes in 3D SPF modeling. As a "chicken-and-egg" problem, it

is difficult to determine the exact gradient magnitude through the geological measurements or prior structural knowledge. We proposed an iterative optimization method which alternates between estimation of SPF and gradient magnitudes so that the gradient magnitudes progressively converge towards the values being adaptive to the stratigraphic architecture. The optimized gradient magnitudes more accurately simulate the variations of the SPF between the top and bottom surfaces.

Jessell et al. (2014) highlighted two limitations of current implicit modeling schemes: (1) they are incapable of interpolating

or extrapolating a fold series within a continuous structural style; (2) the shape of fold hinges they produce is not controlled and may yield inconsistent geometries. To overcome these two limitations, we adopted two strategies: (1) a 3D stratigraphic potential field modeling method based on HRBF interpolant was used to interpolate or extrapolate a fold series within a structurally continuous domain; (2) a number of structural attitude points were sampled on both limbs of the folds to control the geometries of fold hinges.

There are several choices for the value of the potential field, e.g., the sorted serial number or depositional time for each stratigraphic interface (Mallet, 2004). However, the thickness of the stratum is not necessarily proportional to the sorted serial number and deposition time. We chose the burial depth of each stratigraphic interface relative to the top surface of the Quaternary as the SPF value so that stratigraphic thickness is considered as a constraint in the modeling. Compared with using the sorted serial number or depositional age of stratigraphic interfaces as the potential field value, our solution is more in line

with 3D SPF modeling. We derived the gradient direction from the attitude points; moreover, we used the gradient magnitude as a constraint to control the rate of change of the SPF.

**7 Conclusions**

The purpose of this study is to establish a framework for 3D SPF modeling by using the HRBF interpolant with adaptive gradient optimization constrained by on-contact attribute points and off-contact structural attitude points. We applied this

method to a study site in the Lingnian-Ningping area, and a geological map, 4 cross-sections, and a DEM were used as original data to model a SPF whose field value was taken from the relative burial depth of the stratigraphic interfaces. The results show that the implicit modeling of the SPF by HRBF interpolant and optimization of gradient magnitude can be effectively adapted to 3D geological modeling using the sampling points from a geological map and cross-sections. A SPF can express the parameters of a stratum such as property, shape and topology in 3D space. Because 3D stratigraphic potential fields can be

coupled with various geoscience numerical simulation methods, they have a broad prospect for application in related fields such as metallogenic prediction.

However, the modeling process is complicated because the sub-domains are required to be divided manually. In actual geological surveys, the geological structure may be more complex and include a large number of faults, unconformable strata



and intrusive rocks. Therefore, it is necessary to separately identify the boundary of the sub-domains according to the fault interfaces, unconformable strata and intrusive rocks before the 3D geological modeling work. A goal for future work is to introduce a drift function in the model to accommodate discontinuity of fault planes. In addition, the uncertainty of the model should be considered in the modeling process, and additional geophysical exploration data and geological interpretation should be incorporated into the modeling constraints.

*Code availability.*

The source code for the AdaHRBF is available in MATLAB at Github (https://github.com/csugeo3d/AdaHRBF, DOI: https://doi.org/10.5281/zenodo.7340093, Zhang et al., 2022).

*Author contributions.*

Baoyi Zhang and Hao Deng initiated the conception of the study and advised the research on it. Linze Du and Yongqiang Tong programmed the AdaHRBF code and carried out the data analyses for real-world case studies. Umair Khan contributed
significantly to analysis and manuscript preparation. Yongqiang Tong performed both verification and real-world experiments, created all plots, carried out the initial analysis and wrote the manuscript. Hao Deng and Lifang Wang helped perform the analysis with constructive discussions. All authors provided critical feedback and helped to shape the whole study.

*Competing interests.*

No competing interests are present.



*Acknowledgements.* This study was supported by grants from the National Natural Science Foundation of China (Grant Nos. 42072326 and 41972309), China Geological Survey Project (Grant No. DD20190156), and the National Key Research and Development Program of China (Grant No. 2019YFC1805905). The authors thank the MapGIS Laboratory Co-Constructed by the National Engineering Research Center for Geographic Information System of China and Central South University for providing MapGIS® software (Wuhan Zondy Cyber-Tech Co. Ltd., Wuhan, China). We also thank Research Professor ZHOU

Shang-guo (Institute of Mineral Resources Research, China Metallurgical Geology Bureau) and Professor MAO Xian-cheng (Central South University) for their kind assistance with data collection and Professor Jeffrey Dick (Central South University) for revising scientific English writing of this manuscript.

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
