# Peer review of "AdaHRBF v1.0: Gradient-Adaptive Hermite-Birkhoff Radial Basis Function Interpolants for Three-dimensional Stratigraphic Implicit Modeling"

_EGUsphere, 2022_

## Author Comment (AC1)

Dear Dr. Italo Goncalves:

Thanks for your effort to review our manuscript titled " AdaHRBF v1.0: Gradient-Adaptive Hermite-Birkhoff Radial Basis Function Interpolants for Three-dimensional Stratigraphic Implicit Modeling ", and now we have just revised this manuscript according to your good suggestions. The details are as follows, and all the revisions are done using track changes in Word.

RC1: **'Comment on egusphere-2022-1304'**, Italo Goncalves, 17 Feb 2023

In this work, the authors introduce an iterative procedure to deal with the uncertainty in the gradient magnitudes of attitude data in implicit geological modeling, which represent local inverse thickness. The results are very solid, and I believe the manuscript is suitable for publication after the corrections pointed below.

Response: We appreciate your positive comments and grateful for that, and now we have just revised this manuscript according to your good suggestions.

Line 175: if the cubic function is the only one used, I see no reason the present other types in a table. You can point to a reference that lists them and save space. Also, in 3 dimensions the cubic function is the one that minimizes the curvature (eq. 1), the others do not necessarily do so. See Chapter 6 in Rasmussen and Williams (2006) and Wendland (2005).

Response: Thank you for providing the excellent references, we have cited them for listing other radial basis functions and removed Table 1 to save space. We also explained the reason why we use the cubic function as it satisfies Equation 1.

Line 241: isn't there a risk of obtaining a negative $\lambda_{t+1}$ with eq. 8? Have you tried using something like $\lambda_{t+1} = (l_t - l_{t-1})^2$? Also, I suppose the same update is applied to the 3 directions, but it is important to emphasize this.

Response: Thank you for excellent suggestion. Yes, there is a risk of obtaining a small negative $\lambda_{t+1}$ with Eq. 8. We will try $\lambda_{t+1} = (l_t - l_{t-1})^2$, therefore, Eq. 8 and Eq. 9 should converge simultaneously. We have emphasized that we apply the same $\lambda_{t+1}$ to three directions.

If contact data is somehow unavailable or unreliable in part of the space, would the model be able to benefit from off-contact point data (such as indicators for stratum A, stratum B, etc.)? They could be useful to improve the classification accuracy close to the DTM where this information is available. See Hillier et al. (2014).

Response: We do not integrate the off-contact point indicating a lithologic marker in our linear AdaHRBF system. However, the data source of our method mainly includes

geological map and cross-section. In geological mapping, geologist will consider this kind of point when connecting the boundary of a stratum.

This might already be published elsewhere, but I could not see the difference between HRBF and basic RBF. The equations presented seem the same as the basic RBF equations seen in the books. Please clarify the difference and point to the reference that introduced HRBF.

Response: Thank you for the suggestion. Revised.
Generally, the basic RBF reconstructs an implicit function with constraint $f(\boldsymbol{p}_i) = f_i$, however, the HRBF reconstruct an implicit function which interpolates scattered multivariate Hermite-Birkhoff data (i.e., unstructured points and orientations). See Macedo et al. (2011).

A few points to improve the discussion. Please elaborate on how the present work compares to these (which have already been cited):

- This approach is very similar to von Harten et al. (2021), with the difference that here the diagonal matrix is applied to the gradients instead of the contacts.

- Gonçalves et al. (2017) use the strike and dip vectors as zero-gradient directions in order to avoid assigning an arbitrary magnitude to the normals. Have you tried this approach?

- By extending the inequality constraints by Hillier et al. (2014) to the gradients, it is possible to obtain the same results presented here, in principle.

Response: Thank you for the suggestion. Revised.
(1) Because the diagonal matrix directly relates to the input data, von Harten et al. (2021) add variation $\sigma^2$ to the contacts in the contact diagonal matrix to realize the local smoothing as well as we add optimal item $\lambda$ on the gradient diagonal matrix to iteratively get the optimized gradient magnitude.
(2) Your method used a novel zero-magnitude gradient to avoid assigning a magnitude or modulus, which is the tangent constraint. However, we emphasize on iteratively getting the optimized gradient to represent changing trend of stratigraphic potential field.
(3) Besides constraints of scattered multivariate Hermite-Birkhoff data, the Generalized RBF, proposed by Hiller et al. (2014), reconstructs an implicit function with more constraints of lithologic markers (inequality) and lineations (tangent). A goal for future work is to integrate these constraints in our solution to utilize more kinds of modeling data. In our method, the gradients are transformed from off-contact or on-contact structural, and we could not connect the gradient with the inequality, see Fig.1 of Hiller et al. (2014).

I did not find the STI to be very informative of the stratigraphic characteristics of the strata. It seems to be a little erratic and can vary from the minimum to maximum value within the same stratum, which seems to defeat its very purpose. Perhaps trying to assign a geological meaning to the gradients is not a good idea, as they can be very dependent of the specific data points that were used and are a result of the minimum-curvature characteristic of RBF. I think the manuscript would not suffer with the removal of this section.

Response: Thank you for the suggestion. We have removed the contents related to STI.

Minor points:

The manuscript seems to suffer from a compilation error. See pages 6 and 7. All the R symbols are displaced.

Response: We have corrected the compilation error of pdf file.

Lines 59, 348: "true" gradient magnitudes seem a rather strong term. I would call it an optimized gradient.

Response: Thank you for the suggestion. Revised.

Line 184: a line break after the semicolon would improve readability.

Response: Thank you for the suggestion. Revised.

Line 200: it might be worth mentioning that $\theta_2 = \theta_1 + 90°$.

Response: Thank you for the suggestion. Revised.

Figure 2: if the vectors g and n have the same direction but not necessarily the same magnitude, I think the figure could be improved by adding a vector n with a different length.

Response: Thank you for your good suggestion. We have updated Figure 2.

Line 212: please rephrase to avoid starting a new section with "however".

Response: Thank you for the suggestion. Revised.

Line 218: suppress "the".

Response: Revised.

Figure 3: was it hand-drawn or computed? A computed example might make the point clearer.

Response: Thank you for your good suggestion. We have removed the hand-drawn Figure 3 and provided two computed examples (new Figures 4 & 5) to show the inconsistent of SPF caused by forcing equal gradient magnitude for each attitude point.

Line 231: it is worth emphasizing that n is a unit vector.

Response: Thank you for the suggestion. Revised.

Figure 7: I think this example is unnecessary given the previous two, but I leave it at the authors' discretion.

Response: Thank you for the suggestion. We have removed the Figure 7.

Figure 12: is this field value the burial depth that was mentioned before? How was it measured? Is it constant within a given contact?

Response: Yes, the relative burial depth was mentioned in new Figure 8 (Comprehensive stratigraphic column of the study area) from regional geological studies. We assigned an approximate constant for each contact in the whole study area. We have also updated the legend of new Figure 10 using contact points with different colors.

Line 349: "The changes of gradient magnitude **are** shown…"

Response: Revised.

Line 475: "attitude"

Response: Revised.

References:

Rasmussen, C. E., & Williams, C. K. I. (2006). Gaussian processes for machine learning. MIT Press. https://doi.org/10.1142/S0129065704001899

Wendland, H. (2005). Scattered Data Approximation. Cambridge University Press.

Response: Thank you for providing the excellent references, we have cited them for listing other radial basis functions.

---

## Author Comment (AC3)

Dear Dr. Lachlan Grose:

  Thanks for your effort to review our manuscript titled " AdaHRBF v1.0: Gradient-Adaptive Hermite-Birkhoff Radial Basis Function Interpolants for Three-dimensional Stratigraphic Implicit Modeling ", and now we have just revised this manuscript according to your good suggestions. The details are as follows, and all the revisions are done using track changes in Word.

RC2: **'Comment on egusphere-2022-1304'**, Lachlan Grose, 20 Feb 2023

The paper presents AdaHRBF (adaptive hermite-birkhoff radial basis function), a new interpolation method for building implicit geological models. The main contribution of this the iterative process for adapting the gradient of the implicit function to prevent artefacts due to inconsistent gradient magnitude norms.

The paper is generally well written with a logical structure and I believe that it is a good contribution towards the field. In general, the authors have referenced some of the appropriate literature however I believe a deeper analysis of the different implicit modelling techniques would improve the paper – for instance when reviewing discrete smooth interpolation that papers by Mallet 1980/1992 were presented as implicit methods where these papers actually discuss the method applied to 2D surfaces.

Response: We appreciate your positive comments and grateful for that. We thought DSI interpolant could also be applied to implicit modeling with a tetrahedral mesh. We have removed these references to avoid misunderstanding.

The authors should thoroughly review the paper "Three-Dimensional Modelling of Geological Surfaces Using Generalized Interpolation with Radial BasisFunctions" as there are a lot of parallels with the presented works that seem to be missed.

If the method is presented as an approach to tackle the issues with fold geometries it would be worthwhile reviewing the relevant literature around fold modelling: Laurent et al., 2016, Grose et al., 2017,2019, 2020, Hillier et al., 2014.

Response: Thank you for providing the excellent references. We have cited these articles in the Sections Related Works and Discussions.

Don't change between strike and dip data and attitude data, keep it consistent. Preferably something meaningful for geologists.

Response: Thank you for the suggestion. We have changed attitude data to strike and dip data.

What is wrong with the formatting of the pages with equations? All of the equations need to be carefully checked to ensure that they are readable and there are no extra symbols!

Response: We have corrected this compilation error from word file to pdf file.

The figure captions are brief and difficult to follow. They should be stand alone and provide a description and brief interpretation of the contents.

Response: Thank you for the suggestion. We have updated the captions of Figs 1, 2, new 7, new 8, new 12, new 14, and new 17.

I am not sure what the stratigraphic index adds, it is hard to interpret. For example figure 23 section 14 near D1y there is an odd geometry. What causes this? It is orthogonal to the expected orientation of stratigraphy?

Response: Thank you for the suggestion. We have removed the contents related to STI.

**Major comments:**

From reading the manuscript the main contribution is the ability of the interpolator to adapt to variations in the magnitude of the gradient norm. This is a problem that was discussed by Gautier Laurent in doi: 10.1007/s11004-016-9637-y but his paper does not appear to be referenced. In this paper an iterative approach for updating the gradient norm was presented in a discrete modelling approach. This paper should be discussed and compared with the work presented here, I would strongly encourage the authors to provide a comparison between the two methods.

Response: Thank you for providing this excellent reference. We have discussed this article in the Section Optimization of Gradient Magnitude.

I am not convinced that adding the constant to the diagonal component of the second derivative matrix actually changes the magnitude of the gradient norm. I believe that it will just allow for a larger misfit between the orientation observations and the implicit function – which will have the same result but means that any information in the gradient direction will not be incorporated. If the method is actually just removing outlier data then should the message of the paper be changed to this rather than for adapting the magnitude of the gradient norm?

Response: Introducing optimal term of $\lambda$ into HRBF linear solution will simultaneously cause misfit or gradient direction and changing of gradient magnitude. However, we only update the gradient magnitude in the iterations and do not change gradient direction. Meanwhile, when $\lambda$ is finally close to zero, the HRBF linear solution will satisfy the original gradient direction and iteratively obtained gradient magnitude.

I would be interested in seeing a comparison between this method and a discrete approach where the regularisation contribution can locally change, see LoopStructural paper in GMD for a comparison between discrete interpolation and RBF. I would also be interested in seeing the model without any orientation data and just interpolating from the contact locations and also when constraining the direction of the gradient using tangent constraints.

Response: Thank you for the suggestion. We have discussed regularization contribution in LoopStructural in the Section Optimization of Gradient Magnitude. Meanwhile, we have compared the model interpolated by RBF from the contact locations without any orientation.

**Details comments:**

Line 26: replace significance with importance

Response: Thank you for the suggestion. Revised.

Line 29: delete "and has garnered extensive attention from geologists"

Response: Thank you for the suggestion. Revised.

Line 30: Implicit/explicit definition should refer to the approaches as ways of representing surfaces not as methods for building models

Response: Thank you for the suggestion. Revised.

Line 39-46: A lot of the mentioned studies are not reliant on implicit modelling, you could do the same thing with explicit models. E.g. implicit function is not combined directly with geophysics the model is discretized first which means it could be replaced

with a model defined by explicit surfaces. Same point for uncertainty analysis. I would replace this section with relevant references to implicit modelling not just a list of all studies that use implicit modelling

Response: Thank you for the suggestion. We have removed those irrelevant references, and references of implicit modelling are listed in the Section 2 Related Works.

Line 48: delete "in HRBF method", you can do the same in discrete as well

Response: Thank you for the suggestion. Revised.

Line 60: What was adaHRBF method compared with? It is presented as better than an alternative

Response: Thank you for the suggestion. We have revised that AdaHRBF is compared with HRBF interpolant using constant unit normal gradients and RBF interpolant only using contact locations without orientations.

Line 63: "Distribution of attribute" what do you refer to here it is unclear

Response: Revised. We refer to the attribute of scalar field, i.e., the relative buried depth in our manuscript.

Line 70- 73: Mallet reference is for DSI applied to nodes of triangular surface, do you mean to reference mallet 2004 or frank 2007/ caumon 2013?

Response: Thank you for the suggestion. We thought improved DSI interpolant could then be applied to implicit modeling with a tetrahedral mesh. We have removed Mallet (1989,1992) to avoid misunderstanding. These three references better represent implicit interpolation in GoCAD.

Line 76-80: This paragraph started as discrete modelling and then jumps to rbf methods, perhaps keeping them separate will make it easier for the reader

Response: Thank you for the good suggestion. We have separated Section Related Works into two parts, i.e., discrete interpolants and continuous interpolants.

Line 92-94: Renaudeau and Irakarma are discrete or somewhat discrete methods

Response: Thank you for the suggestion. We have reclassified them into discrete interpolants.

Line 118-121: "Moreover, RBF/HRBF-based methods construct implicit field functions separately for each geological interface and extract the zero value equipotential surfaces to locate the geological interface. Therefore, it is difficult to maintain topological consistency between geological bodies, let alone to represent their internal attributes and structural attitudes."

This is not true, the surfe library by Hillier et al 2014 can do all of these points…

Response: Thank you for the suggestion. We have removed this statement.

Line 126: Reference first sentence of paragraph

Line 126: What do you mean by geological maps? Do you mean the outcrop pattern of contacts?

Response: Yes, we mean the outcrop pattern of contacts on the planar geological maps.

Line 130: Change annotation of f1/f2 to something that can't be confused with faults

Response: Revised.

Line 153: optional? Should be optimal?

Response: Yes, it is "optimal".

Line 162: Can you not change the order of the polynomial trend? Hillier et al can?

Response: As same as Hillier et al (2014) mentioned, the degree of the polynomial is restricted to be at most (m − 1) for CPD functions of order m, whereas for SPD functions there is no restriction.

Line 165: define the meaning of f*

Response: f* is the estimation function of f.

Line 174-178: delete table and reference to other basis functions. If you only include r3 then why introduce the others. You could refer the reader to Hillier et al

Response: Thank you for providing this excellent reference, we have cited Hillier et al (2014) for referring other radial basis functions and removed Table 1 to save space.

Line 179-180: The explanation of the construction of the matrices is not clear, it is not obvious what each component represents. Either leave this information for supplementary material if its not necessary for understanding or add more explanation about the different terms.

Response: Thank you for the suggestion. We have added more explanations about the different terms of matrices.

Line 196: "added into modelling process" add references to all of the work that already does this e.g. hillier et al 2014, caumon 2013 etc

Response: Revised. We have added these references.

Line 212: "it is difficult to obtain the gradient magnitude through any geological observation." Delete

Response: Thank you for the suggestion. Revised.

Line 222: "the same gradient magnitudes" what do the red circles indicate

Response: We have removed the hand-drawn Figure 3 and provided two computed examples (new Figures 4 & 5) to show the inconsistent of SPF caused by forcing equal gradient magnitude for each attitude point.

Line 224: "to the Eq. 4" change to "to Eq. 4"

Response: Revised.

Line 224: "used" replace with "and used"

Response: Revised.

Line 227: There are similarities to this diagonal block to the smoothing parameter in Surfe – this was used in Grose et al., 2020 to show a comparison between smoothing regularisation in rbf to the regularisation used by discrete interpolation

Response: We have discussed regularization contribution in LoopStructural in the Section Optimization of Gradient Magnitude.

Line 238-239: My understanding of adding a constraint to the diagonal of the matrix is it allows for the interpolant to have a larger misfit to the constraint. So this means that by iteratively adjusting the diagonal for specific gradient constraint you are actually changing how well those constraints are honoured by the interpolant which includes not just the gradient magnitude but also the orientation constraint.

Response: Introducing optimal term of $\lambda$ into HRBF linear solution will simultaneously cause misfit or gradient direction and changing of gradient magnitude. However, we only update the gradient magnitude in the iterations and do not change gradient direction. Meanwhile, when $\lambda$ is finally close to zero, the HRBF linear solution will honor the original gradient direction and iteratively obtained gradient magnitude.

Line 247: replace convergency with " when convergence is reached"

Response: Revised.

Line 268: "distribution of attribute and attitude points;"

Are the attribute points constraining the value of the implicit field or are they "interface" points as per calcagno where they set the implicit field to be constant along all points related to a single contact? I don't understand how if the points aren't constraining the value of the implicit field this results in a scalar field with the same range as the original dataset when the gradient norms are unit vectors.

Response: Yes, indeed. The attribute point includes an implicit field value.

Line 330: explain the cross section more

Response: Thank you for the suggestion. We have added more explains of cross-sections.

Line 336: "attitude points"

Personally I don't think attitude points speaks to me as a geologist, could you refer to orientation observations or structural observations. At least make it consistent with the geological map, you have angle of strike and dip vector as the legend

Response: Thank you for the suggestion. We have changed attitude data to strike and dip data.

Line 358: "he" replace with "the"

Response: Revised.

Line 470: "However, existing RBF and HRBF interpolants implicitly reconstruct a single geological interface and extract it as the zero-value equipotential surface."

Not true, read and reference Hillier et al., 2014

Response: Revised. We have removed this paragraph.

Line 472: "Moreover, existing RBF and HRBF interpolants need several independent scalar fields to simulate geological interfaces"

This is also not true, Hillier et al use a single scalar field. You also use several scalar field because you represent each fault block independently. If you see the fault modelling method in Grose et al., 2021, this can be used with Surfe and would allow for a single scalar field for stratigraphy.

Response: Revised. We have removed this paragraph.

Line 484: "they are incapable of interpolating or extrapolating a fold series within a continuous structural style" this point by Jessell 2014 was addressed by a few publications Laurent et al., 2016, and Grose et al., 2017,2018,2019,2020 as well as Hillier et al., 2014

Response: Thank you for providing the excellent references. We have cited these articles in the Sections Related Works and Discussions.

Line 486-489: How have you addressed point 1)? You don't extrapolate a fold series in this manuscript, you interpolate a fold shape from gradient constraints but that is not the same. I would remove this section as it is not consistent with the literature.

Response: Revised. We have removed "extrapolate" from point 1).

Line 490-496: This section makes no sense, needs revisiting.

I don't see using the burial depth as being a new contribution from this paper, it is the same method used by various authors Caumon 2013, Hillier 2014, Grose et al., 2020

Response: Revised. We are explaining the burial depth is more suitable in our solution than other type of potential field value.

Line 505: "Because 3D stratigraphic potential fields can be coupled with various geoscience numerical simulation methods, they have a broad prospect for application in related fields such as metallogenic prediction." Delete or move to discussion, don't introduce a new idea in the conclusion

Response: Thank you for the suggestion. We have removed this sentence from Section Conclusion.

Line 510: "A goal for future work is to introduce a drift function in the model to accommodate discontinuity of fault planes. In addition, the uncertainty of the model should be considered in the modeling process, and additional geophysical exploration data and geological interpretation should be incorporated into the modeling constraints."

Move to discussion, but also please ensure that you reference the limitations of a drift function.

Response: Thank you for providing your excellent reference. Revised as "A goal for future work is to introduce a fault integrating way into the implicit model to accommodate discontinuity of fault planes."

---

## Author Comment (AC4)

Dear Dr. Michal Michalak:

Thanks for your effort to review our manuscript titled " AdaHRBF v1.0: Gradient-Adaptive Hermite-Birkhoff Radial Basis Function Interpolants for Three-dimensional Stratigraphic Implicit Modeling ", and now we have just revised this manuscript according to your good suggestions. The details are as follows, and all the revisions are done using track changes in Word.

**RC3**: 'Comment on egusphere-2022-1304', Michal Michalak, 03 Mar 2023

I decided to review this paper from a more optimization perspective since authors use "optimization" terms very often throughout the manuscript. And my field is rather data science in geology where optimization problems are common. But I'm glad to see proper implicit interpolation guys among reviewers who can better evaluate the contribution to their field.

My general opinion is that in the present form it is difficult to evaluate the contribution because it seems that the key methods are not referenced properly. For example, a textbook about Functional Analysis (pure mathematics) is referenced to support applications of interpolation concepts in geology - an unlikely source of information, where in fact there already exist very specific papers about Hermite-Birkhoff interpolation. Moreover, the authors use standard terms (optimization) in a non-standard context which makes the paper difficult to read: I am looking for an optimization criterion but I can't find it.

Response: We appreciate your review efforts. We thought the Hermite-Birkhoff interpolation in pure mathematics could be referenced. We have replaced a new paper about HRBF applied in geoscience. We have also explained why we regard our solution as an optimization.

A positive note: the paper has a logical structure and a neglible overlap with previous work of the author.

Response: We appreciate your positive comments and grateful for that.

Line 20: What do you mean by optimization? Optimization is usually considered either as minimizing something bad (e.g. misfit function) or maximizing something good (e.g. profit). The Wikipedia definition says: "mathematical optimization is the selection of a best element, with regard to some criterion, from some set of available alternatives". Despite many occurrences of "optimization" throughout the manuscript, I cannot find a criterion that is optimized. Instead, I can hypothesize that by "optimizing" the authors mean learning the true value of something. So I would argue that this research is not about optimization. If the authors do not agree, I would like to see an explanation of:

1) why should we consider the results obtained by authors as optimal, i.e. why any other candidate solution is worse than the results proposed by authors in relation to some criterion

2) a thorough description of methods assumed to give optimal results

As a side note, I was once requested by a reviewer regarding why calculating eigenvectors from an orientation matrix should give optimal results. You can see how it was done in the 4.4.2 section of the below paper (the content rather irrelevant for your paper):

Michalak, M. P., Kuzak, R., Gładki, P., Kulawik, A., & Ge, Y. (2021). Constraining uncertainty of fault orientation using a combinatorial algorithm. Computers and Geosciences, 154, 104777.

(https://doi.org/10.1016/j.cageo.2021.104777) , or here (free access): https://github.com/michalmichalak997/3GeoCombine/blob/master/Michalak_2021_combinatorial _accepted.pdf

Response: Thank you for pointing out this issue. We are sorry for the lack of definition of objective function of the optimization. In our setting, both the potential function $f$ and the gradient magnitudes $\mathbf{l}$ are known. Thus, they are optimized by minimizing an objective function, which leads to a minimization problem as:

$$\min_{f,\mathbf{l}} \sum_{i=1}^{N} (f(\boldsymbol{p}_i) - f_i)^2 + \sum_{j=1}^{M} \left( \frac{\partial f(\boldsymbol{p}_j)}{\partial x} - l_j g_j^x \right)^2 + \left( \frac{\partial f(\boldsymbol{p}_j)}{\partial y} - l_j g_j^y \right)^2 + \left( \frac{\partial f(\boldsymbol{p}_j)}{\partial z} - l_j g_j^z \right)^2$$

$$+ \int_{\mathbf{R}^3} \frac{\partial^2 f(\boldsymbol{p})}{\partial^2 x} + \frac{\partial^2 f(\boldsymbol{p})}{\partial^2 y} + \frac{\partial^2 f(\boldsymbol{p})}{\partial^2 z} + 2\frac{\partial^2 f(\boldsymbol{p})}{\partial x \partial y} + 2\frac{\partial^2 f(\boldsymbol{p})}{\partial y \partial z}$$

$$+ 2\frac{\partial^2 f(\boldsymbol{p})}{\partial z \partial x} dx dy dz$$

Given such a challenging optimization problem, it is intractable to solve it directly using common optimization techniques such as variational approach. Inspired by the well-known iterated conditioned mode method, instead, we devise an iterative scheme to optimize potential function $f$ and the gradient magnitudes $\mathbf{l}$ alternatively. In the revised manuscript, we have presented a thorough description of method to show that the iterative update of gradient magnitudes $\mathbf{l}$ in the current form is equivalent to optimize the objective function. Thus, we consider the resulting gradient magnitudes are optimal.

Line 36: "Speed" - when I build a triangulated surface using 800 points, it doesn't take more than two seconds. But when I try to do a similar thing using interpolation methods in GemPy, it takes really long - so I would argue that speed may not be the best marketing candidate for implicit interpolation methods. Moreover, in cokriging methods it is not enough to add surface points - you need to add 3D orientations. But if it is a subsurface terrain, then how do you get an independent orientation measurement? To sum up, I would like to see a discussion about limitations of implicit methods.

Response: Thank you for the suggestion. Comparing with explicit modeling, implicit modeling has the efficiency advantage of avoiding a lot of workloads of human-computer interaction. The 3D orientations are usually surveyed on the outcrops of strata. However, if it is a totally subsurface terrain, we could model it according to other orientations of its conformable strata with outcrops. In the revised manuscript, we have discussed the limitations of our implicit modeling method in Section Discussions.

Line 48: This is the first occurrence of HRBF in the manuscript so it should be preceded with full name. However, here you point to some weaknesses of HRBF and in line 57 you propose HRBF as your main contribution. I'm confused with this presentation.

Response: Revised. We have given the full name in first occurrence of HRBF. What we propose is AdaHRBF, a gradient-adaptive HRBF framework for SPF modeling. We have highlighted it in the revised manuscript.

Line 57: what is actually Hermite-Birkhoff interpolation? The concept should be explained. In the paper, I can see only one reference (except rather inadequate one about functional analysis) about using Hermite interpolation theory in geology (Wang et al. 2018). I would say that the referenced paper better presents the foundational aspect of the method.

Response: Thank you for the suggestion. Revised.

"Generally, the basic RBF reconstructs an implicit function with constraint $f(\boldsymbol{p}_i) = f_i$, however, the HRBF reconstruct an implicit function which interpolates scattered multivariate Hermite-Birkhoff data (i.e., unstructured points and orientations)"

Lines 155-157: I can see three components of the energy function E (two sums and one integral).

1) What do these components represent and how they can be interpreted?

2) I can see that a textbook about functional analysis is referenced to support the equation (Eq.1). Where exactly in this book did you find information about "minimizing smoothness and unevenness of the energy function"? It seems that it is a general mathematical textbook so I would be surprised to see there notions such as "energy function of stratigraphic potential field" or "degree of unevenness". In fact, I have checked the 1972 edition of the Bachman&Narici book and I could not find such concepts. If you found them in 2000 edition, please provide a scan.

3) can you reference other works where Eq. 1 is used?

Response: Thank you for the suggestion. The first and second components represent the misfit between the estimated values and observed contact points and orientation points, respectively. The third component is the second-order derivative of implicit function to represent smoothness of SPF implicit function. What we do is to minimize the energy function. We have removed this reference to avoid misunderstanding.

Line 511: does your work address the problem of subjectivity in implicit methods mentioned by Grose et al. 2021 (text below)? Please discuss.

"The fundamental reasoning behind our approach is that the subjective constraints that are required to capture the geological features with standard implicit algorithms will be one of the greatest sources of uncertainty in the model." (https://doi.org/10.5194/gmd-14-3915-2021)

Response: Thank you for pointing out this issue. We are suffering the same uncertainty problem from the subjectivity in implicit methods so that additional geophysical exploration data and geological interpretation should be incorporated into the modeling constraints in the future.

Title: I would suggest to change the title so that it presents the main value of the research. As of now, the first part of the title contains some technical terms but in my opinion it should point to the added value for three-dimensional stratigraphic implicit modelling. So if it is optimization, then I would like to see the reflection of optimization in the title.

Response: Thank you for the suggestion. We have changed the title as "AdaHRBF v1.0: Gradient-Adaptive Optimized Hermite-Birkhoff Radial Basis Function Interpolants for Three-dimensional Stratigraphic Implicit Modeling"

---

## Referee Report (RR1)

The authors have made an effort to more clearly explain the reasoning behind their method. This should be appreciated. However, new unexplained concepts (e.g. „well-known iterated conditioned mode method", „well known Duchon's energy") are used to support the main reasoning behind posing and solving their minimization problem. The authors write that the conditioned mode method is „well-known". I would argue that this is something very specific and needs more explanations why it is relevant to the proposed method. Especially that the conditioned mode method has something in common with Markov random fields, a concept that does not occurr in the manuscript. I don't recommend any decision (I marked major revision in the system because I have to select something) because I don't do research in functional analysis (note that the authors now cite a theoretical paper from functional analysis Duchon, 1977 for their well known Duchon's energy). So maybe colleagues doing research strictly in functional analysis could say something more decisive.

More detailed comments are given below.

**1 In the response file, the authors write:**

We thought the Hermite-Birkhoff interpolation in pure mathematics could be referenced.

Any valid source of information can be referenced. Because you deleted the Bachman&Narici book from the referenced publications, you have acknowledged that it was invalid. Thank you. Now you have Duchon, 1977 which seems to pose a new yet similar challenge – why is this publication relevant?

**2 In the response file, the authors write:**

Comparing with explicit modeling, implicit modeling has the efficiency advantage of avoiding a lot of workloads of human-computer interaction. The 3D orientations are usually surveyed on the outcrops of strata.

Since my comment was about comparing triangulation and interpolation methods, I will make a point about the difference in computational time.

Interpolation methods, e.g. kriging (but also in Grose et al. https://doi.org/10.5194/gmd-14-3915-2021), often require solving large systems of linear equations. For example, for kriging equations, the size of the linear systems is proportional to the number of N sampling points. This implies that the CPU time for solving this system on a computer is proportional to $N^2$ (Mallet, J.-L., 2002).

However, the Delaunay triangulation of n points in the plane can be computed in O(nlogn) expected time, so triangulation is faster to calculate.

**Theorem 9.12** *The Delaunay triangulation of a set P of n points in the plane can be computed in $O(n \log n)$ expected time, using $O(n)$ expected storage.*

*Proof.* The correctness of the algorithm follows from the discussion above. As for the storage requirement, we note that only the search structure $\mathcal{D}$ could use more than linear storage. However, every node of $\mathcal{D}$ corresponds to a triangle

**195**

*Fig 1 Scan from de Berg et al.*

References:

- Mallet, J.-L., (2002), Geomodeling, Oxford University Press, p. 510.

- De Berg M, Cheong O, Van Kreveld M, Overmars M (2008) Computational Geometry: Algorithms and Applications, 3rd Ed. Springer

**3 In the response file, the authors write:**

Inspired by the well-known iterated conditioned mode method, instead, we devise an iterative scheme to optimize potential function $f$ and the gradient magnitudes $\mathbf{l}$ alternatively.

The authors write that they are inspired by „conditioned mode method", however I cannot see any citation for this inspiration in the manuscript. The authors cannot assume that geological readers are just familiar with the concept of „conditioned mode method". In fact, I can see the following definition (https://en.wikipedia.org/wiki/Iterated_conditional_modes ):

„In statistics, iterated conditional modes is a deterministic algorithm for obtaining a configuration of a local maximum of the joint probability of a Markov random field. It does this by iteratively maximizing the probability of each variable conditioned on the rest."

You don't use Markov random fields in the manuscript, so why the „conditioned mode method" should be relevant to your method?

**4 In the response file, the authors write:**

In the revised manuscript, we have discussed the limitations of our implicit modeling method in Section Discussions.

However, in the Discussions section, I cannot see any new comments about limitations (e.g. use of data from outcrops discussed in the response file). There are only new comments about future work.

**5 The authors explain the three components of the energy function in the response file but not in the revised manuscript.**

**6 What is actually the „novel optimizing term" ? (Line 57)**

**7 In the response file, the authors write:**

„scattered multivariate Hermite-Birkhoff data (i.e., unstructured points and orientations)"

If the definition for Hermite-Birkhoff data is „unstructured points and orientations", then why not writing only „unstructured points and orientations"? I still don't know why the term „Hermite-Birkhoff" appears in the manuscript.

---

## Author Response (AR2)

Dear Editor and Referees:

Thanks for your effort to review our manuscript titled " AdaHRBF v1.0: Gradient-Adaptive Hermite-Birkhoff Radial Basis Function Interpolants for Three-dimensional Stratigraphic Implicit Modeling", and now we have just revised this manuscript according to your good suggestions. The details are as follows, and all the revisions are done using track changes in Word.

**Report #1: Michalak, Michal**

The authors have made an effort to more clearly explain the reasoning behind their method. This should be appreciated. However, new unexplained concepts (e.g. „well-known iterated conditioned mode method", „well known Duchon's energy") are used to support the main reasoning behind posing and solving their minimization problem. The authors write that the conditioned mode method is „well-known". I would argue that this is something very specific and needs more explanations why it is relevant to the proposed method. Especially that the conditioned mode method has something in common with Markov random fields, a concept that does not occurr in the manuscript. I don't recommend any decision (I marked major revision in the system because I have to select something) because I don't do research in functional analysis (note that the authors now cite a theoretical paper from functional analysis Duchon, 1977 for their well known Duchon's energy). So maybe colleagues doing research strictly in functional analysis could say something more decisive.

Response: We appreciate your effort to review our manuscript and grateful for that. We have addressed your main concerns in our manuscript on response to your detailed comments #1 and #3.

More detailed comments are given below.

**1 In the response file, the authors write:**

We thought the Hermite-Birkhoff interpolation in pure mathematics could be referenced.

Reviewer:

Any valid source of information can be referenced. Because you deleted the Bachman&Narici book from the referenced publications, you have acknowledged that it was invalid. Thank you. Now you have Duchon, 1977 which seems to pose a new yet similar challenge – why is this publication relevant?

Response: Apologies for citing Duchon, 1977. The work of Duchon, 1977, to the best of our knowledge, is the first to define the smooth regularizer for thin-plate interpolant and give its solution. This paper could be also difficult to understand for readers of GMD due to its heavy mathematics. To better explain the smooth regularizer, we replace Duchon, 1997 with two alternative literatures (Wahba, 1990; Walder et al., 2006) that are more relevant to implicit modeling.

*Wahba, G., 1990. Spline models for observational data. Society for industrial and applied mathematics.*

*Walder, C., Schölkopf, B. and Chapelle, O., 2006, April. Implicit surface modelling with a globally regularised basis of compact support. In Computer Graphics Forum (Vol. 25, No. 3, pp. 635-644). Amsterdam: North Holland, 1982-.*
See Lines 157-158.

**2 In the response file, the authors write:**

Comparing with explicit modeling, implicit modeling has the efficiency advantage of avoiding a lot of workloads of human-computer interaction. The 3D orientations are usually surveyed on the outcrops of strata.

Reviewer:

Since my comment was about comparing triangulation and interpolation methods, I will make a point about the difference in computational time.

Interpolation methods, e.g. kriging (but also in Grose et al. https://doi.org/10.5194/gmd-14-3915-2021), often require solving large systems of linear equations. For example, for kriging equations, the size of the linear systems is proportional to the number of N sampling points. This implies that the CPU time for solving this system on a computer is proportional to $N^2$ (Mallet, J.-L., 2002).

However, the Delaunay triangulation of n points in the plane can be computed in $O(n\log n)$ expected time, so triangulation is faster to calculate.

References:

• Mallet, J.-L., (2002), Geomodeling, Oxford University Press, p. 510.

• De Berg M, Cheong O, Van Kreveld M, Overmars M (2008) Computational Geometry: Algorithms and Applications, 3rd Ed. Springer

Response: Thank you for pointing out this issue. We have addressed that "implicit modeling often requires a large solution system of linear equations to consume more computational time than explicit modeling, e.g., the Delaunay triangulation".
See Lines 37-39.

**3 In the response file, the authors write:**

Inspired by the well-known iterated conditioned mode method, instead, we devise an iterative scheme to optimize potential function $f$ and the gradient magnitudes **l** alternatively.

Reviewer:

The authors write that they are inspired by „conditioned mode method", however I cannot see any citation for this inspiration in the manuscript. The authors cannot assume that geological readers are just familiar with the concept of „conditioned mode method". In fact, I can see the following definition (https://en.wikipedia.org/wiki/Iterated_conditional_modes ):

„In statistics, iterated conditional modes is a deterministic algorithm for obtaining a configuration of a local maximum of the joint probability of a Markov random field. It does this by iteratively maximizing the probability of each variable conditioned on the rest."

Reviewer:

You don't use Markov random fields in the manuscript, so why the „conditioned mode method" should be relevant to your method?

Response: Sorry for the lack of citation and the confusing explanation. As you pointed out,

the iterated conditioned mode (ICM) method could be strange and unknown to the readers of Geoscientific Model Development. Thus, we have used the term *alternating optimization*, which would be more straightforward to present our optimization scheme, instead of original term. The alternating optimization is an optimization scheme that alternately updates just some variables at a time rather than update of all variables simultaneously. The ICM is an early and representative example of alternating optimization. In the revised manuscript, we have briefly explained the alternating optimization and cited the literature (Bezdek, J. C., & Hathaway) to show that each alternating optimization is well-suited to our optimization problem.

*Bezdek, J. C., & Hathaway, R. J. (2002). Some notes on alternating optimization. In Advances in Soft Computing—AFSS 2002: 2002 AFSS International Conference on Fuzzy Systems Calcutta, India, February 3–6, 2002 Proceedings (pp. 288-300). Springer Berlin Heidelberg.*

See Lines 238-246.

**4 In the response file, the authors write:**
In the revised manuscript, we have discussed the limitations of our implicit modeling method in Section Discussions.
Reviewer:
However, in the Discussions section, I cannot see any new comments about limitations (e.g. use of data from outcrops discussed in the response file). There are only new comments about future work.

Response: Thank you for pointing out this issue. We have discussed the limitations of sub-domain division by fault and uncertainty introduced by using the orientations from outcrops in the Section Conclusions, and these limitations lead to the future work.
See Lines 511-513.

**5 The authors explain the three components of the energy function in the response file but not in the revised manuscript.**

Response: Thank you for pointing out this issue. We have added these explanations to the newly revised manuscript.
See Lines 163-166.

**6 What is actually the „novel optimizing term" ? (Line 57)**

Response: This optimizing term refers to the diagonal matrix $\mathbf{\Lambda}$ in Equation 10. We have modified this statement in Section Introduction because it is not introduced until Equation 10.
See Line 55.

**7 In the response file, the authors write:**
„scattered multivariate Hermite-Birkhoff data (i.e., unstructured points and orientations)"
Reviewer:
If the definition for Hermite-Birkhoff data is „unstructured points and orientations", then why not writing only „unstructured points and orientations"? I still don't know why the term

„Hermite-Birkhoff" appears in the manuscript.

Response: Thank you for the suggestion. The difference between Hermite-Birkhoff radial basis function (HRBF) and standard radial basis function (RBF) is the presence of orientations. We use the term of Hermite-Birkhoff because of two persons' innovations in interpolating using orientations in functional analysis.

See Lines 44-46.

**Report #2: Goncalves, Italo**

The article seems suitable for publication in its current form.

Response: We appreciate your positive comments and grateful for that.

A final remark: it is important to mention right at the introduction that the difference between Hermite-Birkhoff and standard RBF interpolation is the presence of derivatives.

Response: Thank you for pointing out this issue. Revised.

See Lines 44-46.

Minor remarks:

Line 50: "simulate" does not seem to be the ideal word. Perhaps "models", "generates" or "interpolates".

Response: Thank you for the suggestion. Revised.

See Line 53.

Lines 70-72: Gonçalves et al. (2017) fits into the continuous category of models. If I'm not mistaken, Renaudeau et al. (2019) is a continuous model as well.

Response: Thank you for pointing out this issue. We have categorized your article into continuous models, however, Dr. Lachlan Grose suggested that Renaudeau et al. (2019) is a somewhat discrete model.

See Lines 91-93.

**Report #3: Grose, Lachlan**

The revisions seem to cover all of my comments. One clarification is needed on Line 243 - reference to constant gradient should also reference Frank et al 2007 and Caumon 2013. My comment about DSI was that the 1992 mallet papers do not present the 3d algorithm which is presented in frank 2007, caumon 2013.

Response: We appreciate your positive comments and grateful for that.

We also thank you for pointing out this issue. We have added these references to

methods of constant gradient.
See Lines 226-227.